# Multi-Objective Reinforcement Learning: Convexity, Stationarity and Pareto Optimality

**Haoye Lu, Daniel Herman & Yaoliang Yu**
School of Computer Science, University of Waterloo
Vector Institute
{haoye.lu,d2herman,yaoliang.yu}@uwaterloo.ca

## Abstract

In recent years, single-objective reinforcement learning (SORL) algorithms have received a significant amount of attention and seen some strong results. However, it is generally recognized that many practical problems have intrinsic multi-objective properties that cannot be easily handled by SORL algorithms. Although there have been many multi-objective reinforcement learning (MORL) algorithms proposed, there has been little recent exploration of the fundamental properties of the spaces we are learning in. In this paper, we perform a rigorous analysis of policy induced value functions and use the insights to distinguish three views of Pareto optimality. The results imply the convexity of the induced value function's range for *stationary* policies and suggest that any point of its Pareto front can be achieved by training a policy using linear scalarization (LS). We show the problem that leads to the suboptimal performance of LS can be solved by adding strongly concave terms to the immediate rewards, which motivates us to propose a new vector reward-based Q-learning algorithm, CAPQL. Combined with an actor-critic formulation, our algorithm achieves state-of-the-art performance on multiple MuJoCo tasks in the preference agnostic setting. Furthermore, we empirically show that, in contrast to other LS-based algorithms, our approach is significantly more stable, achieving similar results across various random seeds.

## 1 Introduction

The past decade has seen the rapid development of reinforcement learning (RL) algorithms. Recent breakthroughs in RL have made it possible to develop policies that exceed human-level performance: Atari (Mnih et al., 2015), Dota 2 (OpenAI et al., 2019), etc. Despite their great success, the vast majority of RL algorithms are single-objective based. Although many practical problems can be reduced to a SORL task, there is an increasing recognition that many real-world tasks require us to consider their multi-objective nature (Coello, 2000; Pickett & Barto, 2002; Moffaert & Nowé, 2014; Roijers et al., 2013; Abels et al., 2019; Abdolmaleki et al., 2020; Abdelaziz et al., 2021). There are many works that discuss how to find optimal policies in a multi-objective RL (MORL) problem (Gábor et al., 1998; Pickett & Barto, 2002; Moffaert & Nowé, 2014; Roijers et al., 2013; Yang et al., 2019; Parisi et al., 2016; Mahapatra & Rajan, 2020) or a more general dynamic programming setting (Sobel, 1975; Corley, 1985), but the relationship among various definitions of Pareto optimal policies is hardly discussed. Moreover, there is no rigorous analysis of the range of induced value functions, which has been thought hard to characterize and of irregular shapes (Vamplew et al., 2008; Roijers et al., 2013; Reymond & Nowe, 2019). (Note, similar work has been done for mixed policies, but fundamentally differs from the more common stationary policy that is sought in modern RL.) We hope to give researchers well aligned intuitions about MORL problems that can save effort and accelerate the rate of research in the field; it is to this end that we introduce this paper.[1]

Within this paper, we perform a theoretical analysis of MORL problems with an infinite horizon (rigorous proofs are given in Appx B). After a quick review of the MORL setting and three widely-adopted definitions of Pareto efficiency (PE), we begin our analysis by characterizing the effects

---

[1] Extra literature reviews are given in Appx A.

of policy alterations on the induced value function. We find that single-state policy alterations are insufficient to optimize the induced value function in a MORL setting, but show how it can be done by a multi-state update. We also prove that improving in all states is generally not possible. From here, we show that the range of value functions is convex, which suggests that linear scalarization (LS) is not the bottleneck in finding PE policies. We discuss the deficiencies of existing LS-based algorithms as suggested by our theory and fix them by augmenting the reward function using a strongly convex term. These insights motivate us to propose a new MORL algorithm (CAPQL) which achieves state-of-the-art performance on multiple MoJoCo environments in the preference agnostic setting. An ablation study is performed to understand how augmentation affects the algorithm's performance.

## 2 MULTI-OBJECTIVE RL PROBLEM

To begin, we will do a quick review of MORL problems and the notation we will be using, as well as introduce our definitions of Pareto optimality. Like SORL problems, we consider an agent interacting with an environment. At each step, the agent performs an action based on the current state and the environment returns a reward and the next state. Our setting assumes a vector reward in $\mathbb{R}^d$ and is reduced to a SORL problem if $d = 1$. We model the interaction as a Markov Decision Process (MDP) $(\mathcal{S}, \mathcal{A}, R, P, \gamma)$. As usual, $\mathcal{S}$ and $\mathcal{A}$ are the sets of states and actions, and $\gamma \in (0, 1)$ is the discount factor. Our discussion considers finite $\mathcal{A}$ and $\mathcal{S}$. When the agent takes action $a \in \mathcal{A}$ in state $s \in \mathcal{S}$, the environment gives reward $R(a, s) \in \mathbb{R}^d$ and moves to the next state following the transition probability $P(a, s) \in \Delta^{|\mathcal{S}|}$. In this paper, we consider an infinite-horizon MORL problem and assume bounded rewards. Let $\mathbf{R}(s) = [R(a, s)|a \in \mathcal{A}] \in \mathbb{R}^{d \times |\mathcal{A}|}$ and $\mathbf{P}(s) = [P(a, s)|a \in \mathcal{A}] \in \mathbb{R}^{|\mathcal{S}| \times |\mathcal{A}|}$, $\Pi$ the set of all stationary policies, where $\pi \in \Pi$ maps a state to a distribution over actions. Following the work of Roijers et al. (2013), given $\pi \in \Pi$, the induced value function $V^\pi(s) \in \mathbb{R}^d$ returns the expected sum of discounted reward over the interaction trajectory with the initial state $s$,[2]

$$V^\pi(s) := \mathbb{E}\Big[\sum\nolimits_{t=0}^{\infty} \gamma^t R(a_t, s_t)\Big] \text{ with } s_t \sim P(a_{t-1}, s_{t-1}), a_{t-1} \sim \pi(s_{t-1}), s_0 = s. \quad (1)$$

Let $\mu : \mathcal{S} \to [0, 1]$ be the probability distribution over initial states. The expected value function is:

$$V_\mu^\pi := \mathbb{E}\Big[\sum\nolimits_{t=0}^{\infty} \gamma^t R(a_t, s_t)\Big] \text{ with } s_t \sim P(a_{t-1}, s_{t-1}), a_{t-1} \sim \pi(s_{t-1}) \text{ and } s_0 \sim \mu. \quad (2)$$

That is, $V_\mu^\pi = \mathbb{E}_{s_0 \sim \mu} V^\pi(s_0)$. Let $\mathbf{V}^\pi = [V^\pi(s)|s \in \mathcal{S}] \in \mathbb{R}^{d \times |\mathcal{S}|}$, $\mathbb{V}(s) = \{V^\pi(s)|\pi \in \Pi\}$, $\mathbb{V}_\mu = \{V_\mu^\pi|\pi \in \Pi\}$ and $\mathbb{V} = \{\mathbf{V}^\pi|\pi \in \Pi\}$. The Bellman equation (Bellman, 2003) can be written as:

$$V^\pi(s) = \Big(\mathbf{R}(s) + \gamma \mathbf{V}^\pi \mathbf{P}(s)\Big)\pi(s) \quad \text{for } s \in \mathcal{S}. \quad (3)$$

In RL, we are interested in finding a $\pi$ that maximizes $V^\pi(s)$. When $d = 1$, the regular order defined on $\mathbb{R}$ is adopted, and the optimal policy gives the greatest $V^\pi(s)$. For $d > 1$, we consider the *Pareto order* (PO): for real-valued tensors $\mathbf{u}, \mathbf{v}$ of the same shape, $\mathbf{u} \succeq \mathbf{v}$ if every entry of $\mathbf{u}$ is not less than its counterpart in $\mathbf{v}$.[3] For a set of tensors $\mathcal{C}$ of the same shape, $\mathbf{v} \in \mathcal{C}$ is *Pareto efficient (PE)* if for all $\mathbf{u} \in \mathcal{C}$, either $\mathbf{v} \succeq \mathbf{u}$ or $\mathbf{v} \approx \mathbf{u}$. (A set may have multiple PE elements.) In this paper, we are interested in three types of PE policies that are not carefully distinguished in the existing literature (Roijers et al., 2013; Song et al., 2020; Abdolmaleki et al., 2020).

**Definition 1 (Pareto efficient policies)** *For $s \in \mathcal{S}$ and initial state distribution $\mu$, $\pi \in \Pi$ is **single-state PE (SPE)** if $V^\pi(s)$ is PE in $\mathbb{V}(s)$, and it is **distributed initial state PE (DPE)** if $V_\mu^\pi$ is PE in $\mathbb{V}_\mu$. Likewise, $\pi \in \Pi$ is **aggregate PE (APE)** if $\mathbf{V}^\pi$ is PE in $\mathbb{V}$. Let $\Pi_s^*, \Pi_\mu^*$ and $\Pi^*$ denote the sets of policies that are SPE, DPE and APE, respectively.*

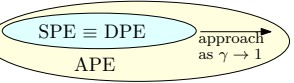

Figure 1: The relationship among three types of PE in Defn 1.

In Sec 4.2, assuming the Markov chain is ergodic, we show that SPE and DPE are equivalent and have the relationship with APE demonstrated in Fig. 1. Since we prove that for all $s \in \mathcal{S}$, $\Pi_s^*$ coincide, we may not specify to which state we are referring to when discussing SPE.

---

[2]The entries of $V^\pi(s)$ are induced value functions in SORL and is known to exist. Thus, $V^\pi(s)$ also exists.

[3]If every entry in $\mathbf{u}$ is strictly greater than its counterpart in $\mathbf{v}$, we write $\mathbf{u} \succ \mathbf{v}$. In general, not all tensors are comparable. For example, given $\mathbf{u} = \binom{1}{0}$ and $\mathbf{v} = \binom{0}{1}$, $\mathbf{u} \not\succeq \mathbf{v}$ and $\mathbf{u} \not\preceq \mathbf{v}$. We write $\mathbf{u} \approx \mathbf{v} \iff (\mathbf{u} \not\succeq \mathbf{v}) \wedge (\mathbf{u} \not\preceq \mathbf{v})$. For single-entry tensors, PO is reduced to the regular order defined on $\mathbb{R}$.

**Optimizing MORL using linear scalarization (LS).**   It is common to convert MORL problems to SORL problems by LS (White, 1993). To do so, take a nonzero vector $\mathbf{w} \in \mathbb{R}^d$ and take the dot product of the reward vector via $\mathbf{w}^\top R(a, s)$. In fact, for any $\pi$, we can left multiply Eq (3) by $\mathbf{w}^\top$ to see that $\mathbf{w}^\top \mathbf{V}^\pi$ is the induced value function of the associated SORL problem. We refer the associated SORL problem with weight $\mathbf{w}$ as $\texttt{SORL}(\mathbf{w})$.

In this paper, we rely on the intimate relationship between MORL and the associated SORL problems to characterize the properties of PE policies (of both types). Our characterization shows that LS does not necessarily inhibit agents from finding desired PE policies, and sheds light on the challenges that modern MORL algorithms face when searching for PE policies.

## 3   FINE-GRAINED CONTROL OF INDUCED VALUE FUNCTIONS

In this section, we study the dynamics of the induced value functions resultant from policy alterations in MORL problems. Our discussion starts by characterizing the effects of single-state policy adjustments. We will show that, unlike SORL, it is not always possible to optimize the value function by optimizing single-state actions. This suggests that the popular Bellman operator, optimizing policies on a single-state basis, does not sufficiently improve a policy's performance in the MORL setting.

The difficulties faced by single-state based optimization techniques reveal the intrinsic difference between the SORL and MORL optimizations, which motivates us to investigate how to adjust multiple states' actions to jointly improve the induced function values.

To make our theoretical results intuitive, we will use the following example throughout this paper.

**Example 1** *Consider a two-objective problem with $\mathcal{S} = \{s_0, s_1\}$ and $\mathcal{A} = \{a_0, a_1\}$. For each state $s$, the chance of staying in $s$ is 50% for both actions. Let $R(a_0, s_0) = [1, 5]^\top$, $R(a_1, s_0) = [5, 1]^\top$, $R(a_0, s_1) = [10, 1]^\top$, $R(a_1, s_1) = [5, 2]^\top$, and $\gamma = 0.5$. The induced value function of $\pi$ satisfies:*

$$V^\pi(s_0) = \left( \begin{bmatrix} 1 & 5 \\ 5 & 1 \end{bmatrix} + \gamma \mathbf{V}^\pi \begin{bmatrix} 0.5 & 0.5 \\ 0.5 & 0.5 \end{bmatrix} \right) \pi(s_0),$$

$$V^\pi(s_1) = \left( \begin{bmatrix} 10 & 5 \\ 1 & 2 \end{bmatrix} + \gamma \mathbf{V}^\pi \begin{bmatrix} 0.5 & 0.5 \\ 0.5 & 0.5 \end{bmatrix} \right) \pi(s_1),$$

*Since $|\mathcal{A}| = 2$, $\lambda_i := \Pr(\text{taking } a_0 \text{ in } s_i)$, $i \in \{0, 1\}$ is sufficient to specify a policy because $\Pr(\text{taking } a_1 \text{ in } s_i) = 1 - \lambda_i$.*

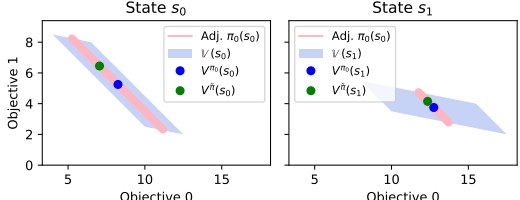

Figure 2: The induced value functions of selected policies in Ex 1. Here, $\pi_0$: $\lambda_0 = \lambda_1 = 0.5$; $\tilde{\pi}$: $\lambda_0 = 0.7$, $\lambda_1 = 0.5$. The pink line segments are the value function range for $\pi \in \Phi(s_0, \pi_0)$, and the light blue patch is the value function range $\mathbb{V}$.

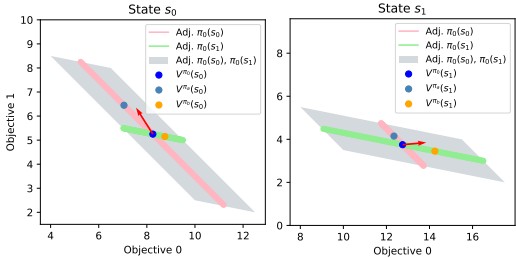

Figure 3: The induced value functions of selected policies in a two-state MORL problem with the configuration given in Ex 1. Here, $\pi_0$: $\lambda_0 = \lambda_1 = 0.5$; $\pi_a$: $\lambda_0 = 0.7$, $\lambda_1 = 0.5$; $\pi_b$: $\lambda_0 = 0.5$, $\lambda_1 = 0.7$. Starting from $\pi_0$, the pink area corresponds to the value functions of $\pi$ obtained by adjusting $\pi_0(s_0)$ (i.e., $\lambda_0$), the green area is the one by adjusting $\pi_0(s_1)$ (i.e., $\lambda_1$) and the grey area is the one by adjusting both. The red vector indicates the shifting direction when increasing $\lambda_0$ and $\lambda_1$ at the same rate.

### 3.1   ADJUSTING THE POLICY IN A SINGLE STATE

In this section, we discuss the properties of the induced value functions when the policy is modified in a single state. In particular, given $s_0 \in \mathcal{S}$ and $\pi_0 \in \Pi$, we temporarily restrict our attention to the policies equal to $\pi_0$ in all states except $s_0$; this is the set:

$$\Phi(s_0, \pi_0) := \{\pi \in \Pi : \pi(s) = \pi_0(s) \text{ for } s \neq s_0\}. \tag{4}$$

Fig 2 plots the value function's range with $\pi_0$ in Ex 1: $\lambda_0 = \lambda_1 = 0.5$. The plot shows, when $\pi(s_0)$ is changed (i.e. $\pi \in \Phi(s_0, \pi_0)$), its induced value function moves along a line segment in both states

with the same moving direction but at different rates. For instance, in Fig 2, we mark the value function of $\tilde{\pi}$ by green dots, where $\tilde{\pi} : \lambda_0 = 0.7, \lambda_1 = 0.5$. In comparison to $\pi_0$ (blue dots), $V^{\tilde{\pi}}$ moves faster in $s_0$ than in $s_1$, which is intuitive because the adjustment is made in $s_0$ and should change its value function by the greatest magnitude. Our observations here indeed hold in general:

**Proposition 1** *For $\pi_0, \pi_1 \in \Phi(s_0, \pi_0)$, let $\pi_\alpha = (1 - \alpha)\pi_0 + \alpha\pi_1$. Then*

$$\frac{\partial V^{\pi_\alpha}(s)}{\partial \alpha} = \frac{\partial V^{\pi_\alpha}(s_0)}{\partial \alpha} \mathbb{E}\left[\gamma^{\tilde{X}(s;s_0)}\right], \tag{5}$$

*where the random variable $\tilde{X}(s; s_0)$ is the number of steps to reach $s_0$ starting from $s$; its distribution is identical for $\pi_0$, $\pi_1$, and $\alpha$. Moreover,*

$$\mathbf{V}^{\pi_\alpha} = (1 - \beta(\alpha; s_0))\mathbf{V}^{\pi_0} + \beta(\alpha; s_0)\mathbf{V}^{\pi_1}, \tag{6}$$

*where $\beta(\alpha; s_0) = \frac{\alpha\phi_1}{(1-\alpha)\phi_0 + \alpha\phi_1}$ and $\phi_i \in [1 - \gamma, 1]$ is a scalar depending on $\pi_i$ and the RL problem settings but independent from $\alpha$.*[4]

In Prop 1, Eq (5) says when $\pi_\alpha(s_0)$ moves from $\pi_0(s_0)$ to $\pi_1(s_0)$, the value functions in all states move along the same direction $\frac{\partial V^{\pi_\alpha}(s_0)}{\partial \alpha}$, while the moving rate is scaled by $\mathbb{E}\left[\gamma^{\tilde{X}(s;s_0)}\right]$ for state $s \in \mathcal{S}$. We note that $\tilde{X}(s_0; s_0) = 0$ and $\tilde{X}(s; s_0) \geq 1$ for all $s \neq s_0$. Therefore, the induced value function always changes the most drastically in $s_0$, which is what we observed in Fig 2.

Eq (6) shows that, if we modify the policy in $s_0$ by letting $\pi_\alpha(s_0)$ be a convex combination of $\pi_0(s_0)$ and $\pi_1(s_0) \in \Phi(s_0, \pi_0)$, then $V^{\pi_\alpha}(s_0)$ is a convex combination of $V^{\pi_0}(s_0)$ and $V^{\pi_1}(s_0)$. Consider the example presented in Fig 2: every policy $\pi' \in \Phi(s_0, \pi_0)$ can be seen as a convex combination between $\pi_0$ and some policy $\pi_1 \in \Phi(s_0, \pi_0)$ with $V^{\pi_1}(s_0)$ on the boundary (where $\pi_1$ depends $\pi'$). Moreover, every point between $V^{\pi_0}(s_0)$ and $V^{\pi_1}(s_0)$ corresponds to a policy in $\Phi(s_0, \pi_0)$.

**Single-state optimization is not sufficient to improve policies in MORL.** It is known that, in SORL problems, a single-state action optimization is sufficient to increase the value function in all states (Sutton & Barto, 2018, p78). Unfortunately, this does not generally hold in MORL problems.

To give a counterexample, consider the setting in Ex 1. Given $\pi_0 : \lambda_0 = \lambda_1 = 0.5$, we plot the induced values of $\pi$ obtained by modifying exactly one state's policy in Fig 3. Here, the pink line segment is the set of value functions obtained by modifying $\pi_0(s_0)$. Similarly, the green line segment is obtained by changing $\pi_0(s_1)$. As we can observe, when updating a single state's policy, one objective must be traded off to optimize the other one. This can be clarified by considering the policy optimization of $s_0$: taking actions $a_0$ and $a_1$ gives the rewards $[1, 5]^\top$ and $[5, 1]^\top$ respectively, so the immediate expected reward for this state is $[-4\lambda_0 + 5, 4\lambda_0 + 1]$. Thus, no single-state policy adjustment will improve both objectives.

### 3.2 ADJUSTING THE POLICY IN MULTIPLE STATES

The insufficiency of optimizing single-state actions motivates us to combine policy adjustments in multiple states to jointly improve the value functions. In Fig 3, the vectors in red give the induced value function's moving directions when we increase $\lambda_0$ and $\lambda_1$ at the same rate. We observe that both objectives are improved in $s_1$, but not in $s_0$. This suggests that we can adjust multiple

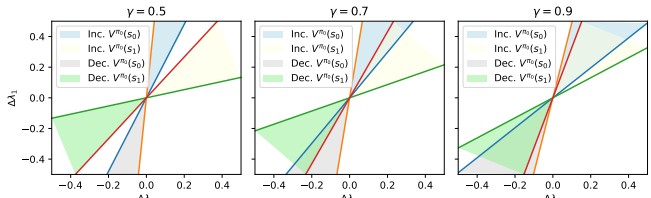

Figure 4: Left: $\gamma = 0.5$. The feasible changes on $\pi_0$ in Ex 1 that increase (decrease) the value function in two states (under the PO). Middle: $\gamma = 0.7$. Right: $\gamma = 0.9$.

states' policies simultaneously to jointly improve its value function in one state. This leads to the natural question: Are there rates of change for $\lambda_0$ and $\lambda_1$ that improves the induced values at both states? The answer is negative. The first graph of Fig 4 plots the feasible changes of $\lambda_0$ and $\lambda_1$ that

---

[4]We give the closed-form expression of $\phi_i$ in Cor 1 in the appendix.

improve $V^{\pi_0}(s_0)$ and $V^{\pi_0}(s_1)$. When $\gamma = 0.5$, the feasible areas (blue and yellow) to improve both objectives in both states are disjoint and hence cannot be improved simultaneously.

The mutual exclusiveness of the two areas is caused by the attenuation effect of the discount factor. Consider the induced value function obtained by increasing $\lambda_0$ from 0.5 to 0.7 for $\pi_0$ (the steel-blue dots in Fig. 3). This adjustment makes the induced function increase in Obj. 1 at the cost of Obj. 0 for both states. Since we are making the adjustment from the perspective of $s_0$, the value function change is greatest in $s_0$ and is dampened by the discount factor $\gamma$ when viewed from $s_1$. Similar observations can be made by increasing $\lambda_1$ from 0.5 to 0.7. This attenuation of the reward propagation makes it impossible for the induced values to move in the same direction in all states, which prevents them from being optimized simultaneously. The last two plots of Fig. 4 show that when $\gamma$ increases, the feasible areas to improve induced values in both states overlap, which corroborate our claims. We generalize our observations by proving:

**Proposition 2** *Assume $\pi_0 \in \Pi$ is not optimal in the associated scalar problem for all $\mathbf{w} \neq \mathbf{0}$. Let $s_0 \in \mathcal{S}$. Then any neighbourhood of $\pi_0$ contains $\pi_1, \pi_2 \in \Pi$ such that for any $\mathbf{u} \in \mathbb{R}^d$ we have*

$$V^{\pi_1}(s_0) - V^{\pi_0}(s_0) = \xi_1 \mathbf{u} \text{ and } \mathrm{V}_\mu^{\pi_2} - \mathrm{V}_\mu^{\pi_0} = \xi_2 \mathbf{u} \quad \text{for some } \xi_1, \xi_2 > 0. \tag{7}$$

*Moreover, if the MDP is ergodic, as $\gamma \to 1$, $V^{\pi_1}(s) - V^{\pi_0}(s) \to \xi_1 \mathbf{u}$ for all $s \in \mathcal{S}$.*

Prop 2 says that, under some weak conditions, we can move the induced value of $\pi_0$ in a state (or the expected induced value) along any direction. Setting $\mathbf{u} = \mathbf{1}$, we improve the induced values. When the MDP is ergodic, as $\gamma \to 1$, the values in all states move along $\mathbf{u}$ and will be optimized at the same time.

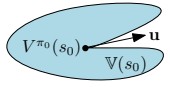

Figure 5: $V^{\pi_0}(s_0)$ is not optimal for any $\mathbf{w}$ but cannot move along $\mathbf{u}$.

**Remark 1** *We note that Prop 2 is non-trivial because a policy's suboptimality over all associated scalar problems does not automatically imply the induced value function can move in any direction. For example, in Fig 5, $V^{\pi_0}(s_0)$ is not optimal in any associated SORL problem but cannot move along $\mathbf{u}$: Prop 2 says $\mathbb{V}(s_0)$ cannot have a shape like this.*

**Remark 2** *The first plot of Fig 4 shows $\pi_0$ cannot be further improved in both states at the same time. This implies $\pi_0$ is in fact APE. However, Fig 2 shows $V^{\pi_0}(s_0)$ is in the middle of the induced value function range. Thus, there exists a policy $\pi$ that has an induced value in $s_0$ greater than $V^{\pi_0}(s_0)$, and hence $\pi_0$ is not SPE in $s_0$. This observation tells us SPE and APE are different in general.*

## 4 CONVEXITY AND PARETO EFFICIENT POLICIES

The analysis of the induced value function's behaviour in Sec 3 provides us with powerful tools to characterize their properties. In this section, we use these tools to show the convexity of the induced function's range and the relationships among the three types of PE.

### 4.1 CONVEXITY OF THE INDUCED FUNCTION RANGES

For more than a decade, it has been considered true that the induced value functions' ranges are irregularly shaped (for stationary policies) (Vamplew et al., 2008; Roijers et al., 2013; Hayes et al., 2022). In fact, this is the key reason behind the belief that LS is not powerful enough to find all PE policies. Prop 3 shows that this belief is not true, and the ranges of the functions are actually convex.

**Proposition 3** *For $s \in \mathcal{S}$, $\mathbb{V}(s)$ is convex. Also, $\mathbb{V}_\mu$ is convex.*

The convexity of $\mathbb{V}(s)$ can be shown by repetitively applying Prop 2 to construct a path between $V^{\pi_0}(s_0)$ and $V^{\pi_1}(s)$ with $\pi_0, \pi_1 \in \Pi$. Roughly speaking, for any point $\mathbf{v}$ over the path, starting from $V^{\pi_0}(s)$, we can repetitively use Prop 2 to construct a sequence of policies with the induced value approaches $\mathbf{v}$. Thus, the path is included in $\mathbb{V}(s)$, and by definition, $\mathbb{V}(s)$ is convex. We can apply the similar idea to show $\mathbb{V}_\mu$ is convex.

**Remark 3** *Prop 3 can also be proved as a corollary of the convexity of the occupancy measure, initially derived in constrained Markov decision theory (Kallenberg, 1983; Puterman, 1994; Altman, 1999). We discuss their relationship and provide a second proof of Prop 3 in Appx C.*

**LS is not a bottleneck in finding PE policies.** The convexity of the induced value functions' ranges suggest that we can potentially find all SPE (DPE) policies through LS. In particular, we have:

**Proposition 4** *For $s \in \mathcal{S}$, $\pi \in \Pi_s^*$ if $\pi$ is optimal in a* SORL($\mathbf{w}$) *with some $\mathbf{w} \succ \mathbf{0}$. Also, if $\pi \in \Pi_s^*$, $V^\pi(s)$ is optimal in a* SORL($\mathbf{w}$) *with a nonzero $\mathbf{w} \succeq \mathbf{0}$.*

As a result, a SPE (DPE) policy achieves optimality in some associated SORL problem. This also suggests that we can potentially find all SPE (DPE) policies by choosing different weights in LS.

**Problems of the existing LS-based algorithms.** We suggest that there are two significant reasons why many algorithms cannot find a rich set of PE policies: determinism and numerical instability.

The majority of RL algorithms favour determinism. While it is well-known that an optimal deterministic policy always exists for SORL problems (Puterman, 1994, Ch 6), almost all PE value functions for a MORL problem require stochastic policies. In Fig. 6, we plot the value functions of state $s_0$ for different policies in Ex 1. We see that most PE policies are stochastic (squares) while the deterministic ones can only cover the vertexes (stars). This observation shows that *unless we ensure current algorithms can favour stochastic policies, we have implicitly excluded almost all PE policies.* Additionally, it is numerically impossible to obtain some SPE policies in practice although they can be found in theory. In fact, almost all nonzero weights $\mathbf{w} \succeq \mathbf{0}$ correspond to the SPE policies on the vertexes. For instance, in Fig. 6, we show the choice of weights for finding a specific SPE policy in the up-right corner. While there is a wide range of weight selections for finding the SPE policies on the vertexes, only those on the boundary of the green-cyan/cyan-purple patches can be used to find the stochastic policies upper/right boundary of the polytope. Specifically, only weight vectors normal to the Pareto front can be used to find stochastic policies. In practice, we do not know how to find such weights; even if we do, they cannot be picked as a tiny perturbation over them gives a weight corresponding to a vertex. Besides, these weights are shared by a set of stochastic SPE policies (on the same facet of the polytope). Thus, the algorithm would produce a random policy in the set with an undesired value function even if the right weight is picked.

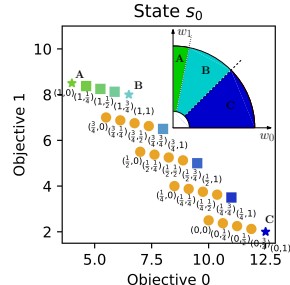

Figure 6: Value functions of $s_0$ for various $(\lambda_0, \lambda_1)$ in Ex 1. The stars are deterministic SPE policies and the squares are stochastic ones; sub-optimal ones are marked by golden dots. The choice of weights for finding different SPE policies are plotted in the top-right corner, where the patch colours correspond to the found SPE policies'.

**Adding a strongly concave term fixes the problems.** Let $f : \Delta^{|\mathcal{A}|} \to \mathbb{R}$ be a strongly concave function for action-taking distributions. We then replace the regular reward $R(a_t, s_t)$ with $\tilde{R}(a_t, s_t) = R(a_t, s_t) + \alpha f\big(\pi'(s_t)\big)\mathbf{1}$, where $\mathbf{1} \in \mathbb{R}^d$ is a vector of ones and $\alpha > 0$ controls the strength of the augmentation effect. Then the induced value function of a policy $\pi$ for initial state $s \in \mathcal{S}$ under this augmented setting becomes:

$$V_{\alpha f}^\pi(s) := \mathbb{E}\Big[\sum_{t=0}^\infty \gamma^t \Big(R(a_t, s_t) + \alpha f\big(\pi(s_t)\big)\mathbf{1}\Big)\Big] \text{ with } s_t \sim P(a_{t-1}, s_{t-1}), a_{t-1} \sim \pi(s_{t-1}), s_0 = s. \quad (8)$$

Let $\mathbb{V}_{\alpha f}(s) = \{V_{\alpha f}^\pi(s) | \pi \in \Pi\}$, $\mathbb{W}^+ = \{\mathbf{w} \succeq \mathbf{0} | r_1 \le \|\mathbf{w}\|_1 \le r_2\}$ for some $r_1, r_2 > 0$ and $\mathbb{V}_{\alpha f}^\star(s)$ the set of PE elements in $\mathbb{V}_{\alpha f}(s)$.

Fig 7 shows how the reward augmentation changes the value function's range. We observe that the augmentation makes the shape of the PE element set $\mathbb{V}_{\alpha f}^\star(s)$ strictly convex. Thus, for every $\mathbf{w} \in \mathbb{W}^+$, there is a unique $V_{\alpha f}^\pi(s) \in \mathbb{V}_{\alpha f}^\star(s)$ that has the maximum projection on $\mathbf{w}$. Let $g_s$ denote this unique correspondence relationship from $\mathbb{W}^+$ to $\mathbb{V}_{\alpha f}^\star(s)$:

$$g_s(\mathbf{w}) = \text{argmax}_{\mathbf{v} \in \mathbb{V}_{\alpha f}^\star(s)} \mathbf{w}^\top \mathbf{v}. \quad (9)$$

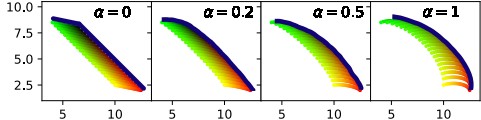

Figure 7: The effects on the induced value functions at $s_0$ for selected policies in Ex 1 by adding strongly concave terms to the immediate rewards with different alpha. Here, $f$ returns the entropy of action taking distribution. $\mathbb{V}_{\alpha f}^\star(s_0)$ is marked in blue and the dots of the same colour among the four plots correspond to the same policy.

From Fig 7, we can also observe that, the strict convexity of $\mathbb{V}^\star_{\alpha f}(s)$ makes $g_s$ (uniformly) continuous. As we rotate $\mathbf{w}$ clockwise, $g_s(\mathbf{w})$ slides from the left end of the blue curve the right for last three plots; in contrast, for the first one, it will jump from top left vertex to the middle one followed by the bottom right one. The continuity of $g_s$ makes it numerically possible to pick a good $\mathbf{w}$ with $g_s(\mathbf{w})$ close to the desired PE element in $\mathbb{V}^\star_{\alpha f}(s)$. We summarize our observations in Prop 5.

**Proposition 5** *Function $g_s$ given in* (9) *is well-defined, surjective and uniformly continuous.*

Finally, the strongly concave term naturally injects the preference on the stochastic policies. Therefore, all the problems of the existing LS-based methods mentioned in Sec 4.1 has been fixed.

**Remark 4** *Function $g_s$ corresponds to the extended target value function considered by Abels et al. (2019) and Yang et al. (2019). In their work, they implemented an extended Q-network, $Q(s, a, \mathbf{w})$ and minimize $\|g_s(\mathbf{w}) - \mathbb{E}_a[Q(s, a, \mathbf{w})]\|_2$ for all $s \in \mathcal{S}, a \in \mathcal{A}$ and $\mathbf{w} \in \Phi \subseteq \mathbb{W}^+$. In Sec 5, we will see that, as they adopted regular rewards without augmentation, the target value function $g_s$ is sensitive to the input $\mathbf{w}$, making the training process unstable and lowering the algorithms' performance (Appx D describes the causes of this training instability). We will instead propose a concave-augmented Pareto Q-learning algorithm (CAPQL). Our empirical study shows that the algorithm's performance improves with significantly more stable training trajectories.*

**Remark 5** *Fig 7 suggests that a smaller $\alpha$ preserves more information about the original problem but makes $g_s(\cdot)$ less robust to perturbations and lowers the stability of the algorithm approximating $g_s$. Likewise, a larger $\alpha$ improves the stability but would harm the algorithm's expected performance.*

## 4.2 RELATIONSHIPS AMONG THREE TYPES OF PE

Before we introduce our new LS-based MORL algorithm, we will complete our theoretical discussion about PE. In particular, we prove that the three types of PE have the relationship summarized in Fig. 1. Throughout this section, we assume that the MDP is ergodic.

**SPE is state-independent and equivalent to DPE.** According to Prop 4, if $\pi \in \Pi^*_s$ for $s \in \mathcal{S}$, then it must be the optimal policy for some associated scalar problem with some nonzero $\mathbf{w} \succeq \mathbf{0}$. When $\mathbf{w} \succ \mathbf{0}$, we apply Prop 4 to conclude that $\pi \in \Pi^*_{s'}$ for all $s' \in \mathcal{S}$ and $\pi \in \Pi^*_\mu$. If $\mathbf{w}$ contains zero entries, the problem can be approximated by assigning an arbitrarily small positive weight to the zero entries. Then, the problem is reduced to case with $\mathbf{w} \succ \mathbf{0}$. [5] Since $\pi$ is optimal in the associated problem of all states, none of its value functions can be improved independently or in aggregation. Hence, $\pi \in \Pi^*$ as well. We summarize the derived results in:

**Proposition 6** *If $\pi \in \Pi^*_s$ for some state $s \in \mathcal{S}$, then for all $s' \in \mathcal{S}$, $\pi \in \Pi^*_{s'}$. Thus, the sets of single-state PE policies coincide for all initial states and is a subset of $\Pi^*$.*

Using a similar derivation by linking the PE policies to the optimal solutions of the corresponding associated scalar problem also proves:

**Proposition 7** *A policy is SPE if and only if it is DPE. That is, for $s \in \mathcal{S}$, $\Pi^*_s = \Pi^*_\mu$.*

**SPE implies APE but not vice versa in general.** For a selected state $s \in \mathcal{S}$, a policy $\pi$ is SPE as long as its induced value function $V^\pi(s)$ is PE in $\mathbb{V}(s)$. The function values for other states are not relevant. In contrast, APE involves the induced function value in all states. A policy $\pi$ is APE if we have to trade off the value functions of some states for improving those of the others.

For ergodic SORL problems, the induced value function of a policy $\pi$ reaches the maximum in one state if and only if it reaches the maximum in all states (Prop 2.1.2 in (Bertsekas, 2022)). This implies that SPE and APE are equivalent in a SORL problem. However, the equivalence does not generally hold for MORL problems. In Rmk 2, we presented a case where $\pi_0 \in \Pi^*$ but $\pi_0 \notin \Pi^*_{s_0}$. Besides, Prop 6 shows that if $\pi \in \Pi^*_s$ for some $s \in \mathcal{S}$, then $\pi \in \Pi^*$. As a result, $\Pi^*_s \subset \Pi^*$ but do not necessarily coincide.

---

[5]We provide a more rigorous proof in the appendix by constructing a lexicographic order on $\mathbb{R}^d$.

As we have noted in Sec 3.2, this proper subset relationship is due to the attenuation of the reward propagation caused by the discount factor. The attenuation makes the moving direction of the value function differ among various states and causes the feasible sets of improvement adjustments to be disjoint (see Fig 4). When $\gamma \to 1$, the moving directions of all states converge (Prop 2). Therefore, the feasible sets of improvement adjustments for different states will also converge and eventually overlaps for sufficiently large $\gamma$ (see the last two plots of Fig 4). Therefore, improvements can be made in all states until they reach the boundaries of the value functions' ranges. In other words:

**Proposition 8** *For all $s \in \mathcal{S}$, as $\gamma \to 1$, the SPE policy set $\Pi_s^*$ approaches the APE one $\Pi^*$.*

## 5 CONCAVE-AUGMENTED PARETO Q-LEARNING

Motivated by the discussions in Sec 4.1, in this section, we develop a new Q-learning algorithm with the reward augmented by a strongly concave term. We call our new algorithm *concave-augmented Pareto Q-learning* (CAPQL).

### 5.1 MORL PROBLEM WITH AGNOSTIC WEIGHT PREFERENCE

Our CAPQL algorithm is designed for solving MORL problems where the preference weights for the objectives are (potentially) different between episodes and are not known in advance. The setting was initially considered by Abels et al. (2019) to propose a multi-objective Q-network (MOQ). In particular, the problem considers a set of weights $\mathbf{\Phi} \subseteq \mathbb{W}^+$. For each episode, a preference weight $\mathbf{w} \in \mathbf{\Phi}$ is given, and the algorithm is expected to maximize the sum of the rewards projected onto $\mathbf{w}$. Thus, the algorithm has to handle all possible weights in $\mathbf{\Phi}$.

### 5.2 IMPLEMENTATION OF CAPQL

Following Abels el al.'s work, we consider an extended Q-network, $Q(s, a, \mathbf{w})$, and train it to approximate the Q-values of a optimal policy of $\text{SORL}(\mathbf{w})$ for all $\mathbf{w} \in \mathbf{\Phi}$. Unlike MOQ that uses the reward $R(a_t, s_t)$ from the environment directly, CAPQL replaces it with $\tilde{R}(a_t, s_t) = R(a_t, s_t) + \alpha f\big(\pi'(s_t; \mathbf{w})\big)\mathbf{1}$, where $\alpha > 0$.

Notice that scaling $\mathbf{w}$ does change the selection of the induced value function $V_{\alpha f}(s)$ that has the greatest projection on it. Without loss of generality, we assume $\|\mathbf{w}\|_1 = 1$. Furthermore, we set $f$ to be the entropy operator $\mathcal{H}(q) = -\int q(a) \log q(a)\,\mathrm{d}a$ (discussions on the selection of $f$ are given in Appx E). We train our algorithm by optimizing it over $\text{SORL}(\mathbf{w})$, for all $\mathbf{w} \in \Phi$. For a fixed $\mathbf{w}$, the learning task of $\text{SORL}(\mathbf{w})$ can be written as

$$\pi(\cdot\,; \mathbf{w}) = \underset{\pi'(\,\cdot\,; \mathbf{w})}{\operatorname{argmax}} \ \mathbb{E}\Big[ \sum\nolimits_{t=0}^{\infty} \gamma^t \ \big( \mathbf{w}^\top R(a_t, s_t) + \alpha \mathcal{H}\big(\pi'(s_t; \mathbf{w})\big) \big) \Big], \tag{10}$$

which is obtained by projecting the value function defined in (8) onto $\mathbf{w}$ followed by taking the maximum over the policies conditioned on it. Interestingly, this is the MORL extension of the learning task considered in SAC (Haarnoja et al., 2018). Hence, we implement the algorithm with the Q-network and policy network conditioned on $\mathbf{w}$. In each training step, we first sample a weight $\mathbf{w}'$ and follow the SAC method to train the policy and the Q-network conditioned on it. (The implementation details, pseudocode are given in Appx F and convergence property is discussed in Appx G.)

As discussed in Prop 5 and Rmk 4, the optimal policy $\pi$ of $\text{SORL}(\mathbf{w})$ has the value function $g_s(\mathbf{w})$. Thus, the training is to make $\mathbb{E}_{a \sim \pi(s;\mathbf{w})}[Q(s, a, \mathbf{w})]$ match $g_s(\mathbf{w})$ for all $s \in \mathcal{S}$ and $\mathbf{w} \in \mathbf{\Phi}$. As Prop 5 shows, adding the entropy term makes the target $g_s(\mathbf{w})$ uniformly continuous with respect to $\mathbf{w}$, which is easier to fit and is less numerically unstable (in Appx H, we empirically show this phenomenon by training the algorithm over Ex 1 and visualizing its $g_s(\mathbf{w})$). Hence, we should expect CAPQL to both converge faster and be more numerically stable during training.

### 5.3 EXPERIMENTS

We test our algorithm over a multi-objective version of the MuJoCo environment. The reward vector was created by simply exposing the individual components that went into the regular scalar reward:

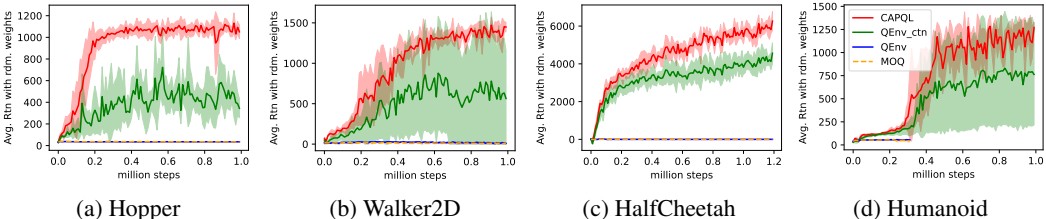

Figure 8: Training curves of the MORL algorithms in MuJoCo environments with vector rewards.

adding them up recovers the default scalar reward. (See Appx I-Table 4 for details.) We restrict $\mathbf{\Phi}$ to only contain weights within 22.5 degrees of the unit vector to ensure that $\mathbf{w} \succ \mathbf{0}$. Finally, we will also perform an ablation study to understand how the algorithm's performance changes with different strength of reward augmentation.

We compare our method to two popular LS-based algorithms: MOQ (Abels et al., 2019) and EnvQ (Yang et al., 2019). MOQ can be seen as a special case of CAPQL as $\alpha \to 0$ and EnvQ is its enhanced version. It has been shown that EnvQ enjoys a higher sampling efficiency and has a consistently better performance than MOQ on multiple MORL benchmarks. Since MOQ and EnvQ were proposed under the finite action setting, to adapt them to MuJoCo's continuous action space, we follow Tang & Agrawal (2020)'s work and discretize each action dimension into five discrete values (e.g., Hopper has three action dimensions; then there are $5^3 = 125$ actions). Additionally, we implement a continuous action version of EnvQ combined with an actor-critic formulation (referred as QEnv-ctn) for comparison. (The model and training configurations are summerized in Appx I).

Fig 8 plots the methods' trajectories on four MuJoCo benchmarks. We train each method five times with various random seeds and report the mean and standard deviation. In every step, we test them over ten randomly sampled weights. We observe that CAPQL has a consistently better performance over all benchmarks and enjoys a faster convergence speed. Additionally, compared to QEnv-ctn, CAPQL has a far more stable training trajectory over different random seeds.

**The relationship between the augmentation strength and CAPQL's performance.** In Rmk 5, we discussed how the augmentation would affect the CAPQL's performance as its strength varies. In Fig 9, we plot the training curves of CAPQL for Hopper with different $\alpha$, which corroborates our claim. In particular, as $\alpha$ increases, the target value function $g_s(\mathbf{w})$ defined in Prop 5 becomes less sensitive to $\mathbf{w}$. Thus, it can be learned more easily; meanwhile, we observe a faster convergence and stabler training trajectories over various random initial seeds. However, if $\alpha$ becomes too large (the case with $\alpha = 0.8$), $g_s(\mathbf{w})$ will significantly deviate from the original one (i.e., when $\alpha = 0$). Then, the algorithm's performance after convergence starts to drop.

Figure 9: Training curves of CAPQL with different $\alpha$ for Hopper.

## 6 DISCUSSION

This paper performed a rigorous analysis of the dynamics of the induced value functions resultant from policy alterations in MORL problems. We analyzed the behaviours of the functions when a single state's policy is altered and showed that this is insufficient to optimize the induced value functions in a MORL setting. We then discussed how to update a policy in multiple states to improve the value function of a specific state. We also showed that when $\gamma \to 1$, the induced values of all states will be improved as well. These insights into the induced value function's properties helped us show the convexity of their range and prove that LS is sufficient to find all SPE (DPE) policies. The equivalence of SPE and DPE was shown, which are also equivalent to APE when $\gamma \to 1$. Next, we showed why existing LS-based algorithms fail and proposed the CAPQL algorithm to address these issues; our empirical evaluation indicates CAPQL's superior performance and corroborates our theoretical analysis.

## REPRODUCIBILITY

**Theoretical Work**    All theoretical results have formal proofs provided in the Appx B.

**Empirical Work**    We provide a detailed pseudocode description of our CAPQL implementation (Alg 1) in Appx F. The parameters used to train and implement all algorithms we analyzed are listed in Tables 1-3 in Appx I. Additionally, we have provided details on the environment configuration and our derived reward functions in Table 4. The source code of our CAPQL implementation is available online: https://github.com/haoyelu/CAPQL.git.

## ACKNOWLEDGEMENT

We thank the reviewers and area chair for constructive comments. We gratefully acknowledge funding support from NSERC and the Canada CIFAR AI Chairs program. Resources used in preparing this research were provided, in part, by the Province of Ontario, the Government of Canada through CIFAR, and companies sponsoring the Vector Institute.

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

## A  RELATED WORK

In recent years, there have been relatively few developments in the abstract analysis of the spaces involved in MORL problems. Much of the previous work done on analyzing the set of induced value functions has been done on a general state-space while allowing various kinds of deterministic or mixed policies (Feinberg & Shwartz, 1995; Vamplew et al., 2008; 2009; Roijers et al., 2013; Barrett & Narayanan, 2008; Moffaert & Nowé, 2014). In particular, Feinberg & Shwartz (1995) found that the expected discounted sum of rewards $V^\pi(s)$ with a fixed initial state $s$ is a convex set when $\pi$ is allowed to be non-stationary. Moreover, they prove that Pareto optimality (at any given state) is equivalent to a linear scalarization (LS) problem under a specific weight vector. Similar convexity and Pareto results were found by Vamplew et al. (2008; 2009), but instead of allowing a general non-stationary policy, they restricted their policies to the set of mixed deterministic policies. The key difference between our work and theirs is that we restrict our analysis to stochastic stationary policies. Hernández-Lerma & Romera (2004) also performed an analysis in a similar setting to us but focused on the feasibility of finding Pareto optimal solutions for specific weights.

Mannor & Shimkin (2004) have also done similar work and required that there exists a common accessible state from every other state; ergodicity is a sufficient condition for their setting. However, their expected sum of rewards is un-discounted, which is not reflective of common modern formulations. They also focused significantly on directional policy optimization and proposed several algorithms that require mixed deterministic policies to function. Apart from the theoretical works, Abels et al. (2019) generalized the Q-function by conditioning it on the importance of the objectives. In their problem setting, the objective preference changes between different episodes and is not known in advance. Therefore, the learning task requires the algorithm to perform well over all potential weight selections. Yang et al. (2019) extended Abels et al.'s work. In particular, they developed a new Bellman optimality operator using envelope updates and proposed a new algorithm for MORL called envelope Q-learning (QEnv). They showed that their algorithm consistently performs better over multiple MORL benchmarks and is more sample efficiency. Finally, Abdolmaleki et al. (2020) have developed an algorithm that allows users to choose objective preferences in a scale-invariant way by restricting the relative influence of each objective when improving the policy.

It should be noted that significant efforts have been made to find alternatives to LS due to perceived and real drawbacks. For example, Van Moffaert et al. (2013) demonstrate that Chebyshev metric can dominate LS in the discrete policy setting since it can be used to find Pareto efficient policies that are in the interior of the convex hull. While one of its primary benefits over LS no longer holds in the stochastic policy setting, it may be interesting to see if an extension of this methodology to our setting could still be beneficial. Alternatively, recent papers in the domain of concave utility reinforcement learning (CURL) have achieved some intriguing results (Geist et al., 2021; Zhang et al., 2020; Agarwal et al., 2022). In the recent work of Agarwal et al. (2022), they seek to find a single policy that maximizes a concave utility function applied to $V^\pi(s)$. While their setting, methods, focuses and insights are largely different from ours, they analyzed their framework's sample efficiency and developed an actor-critic method that reduces the variation of the policy gradients by directly shifting them with a state-dependent term.

## B  PROOFS

Given $s_0 \in \mathcal{S}$ and $\pi_0 \in \Pi$, we temporarily restrict our attention to the policies equal to $\pi_0$ in all states except $s_0$; this is the set:

$$\Phi(s_0, \pi_0) := \{\pi \in \Pi : \pi(s) = \pi_0(s) \text{ for } s \neq s_0\}. \tag{11}$$

**Proposition 9** *Let $\tilde{V}(s; s_0)$ and $\tilde{X}(s; s_0)$ denote the random variables that, starting from $s \in \mathcal{S}$, the sum of the discounted rewards and the number of steps before the first time reaching $s_0$.[6] (The distributions of $\tilde{V}(s; s_0)$ and $\tilde{X}(s; s_0)$ do not depend on $\pi(s)$ and thus are identical for all $\pi \in \Phi(s_0, \pi_0)$.) Write*

$$\mathsf{V}(s_0) = \left[\mathbb{E}\,\tilde{\mathsf{V}}(s'; s_0)\,\text{for } s' \in \mathcal{S}\right] \in \mathbb{R}^{d \times |\mathcal{S}|}, \quad \Gamma(s_0) = \left[\mathbb{E}\,\gamma^{\tilde{\mathsf{X}}(s'; s_0)}\,\text{for } s' \in \mathcal{S}\right] \in \mathbb{R}^{1 \times |\mathcal{S}|}. \tag{12}$$

---

[6]When $s = s_0$, $\tilde{\mathsf{V}}(s_0; s_0) = \tilde{\mathsf{X}}(s_0; s_0) = 0$.

*For $\pi \in \Phi(s_0, \pi_0)$, we have*

$$\mathbf{V}^\pi = \mathsf{V}(s_0) + V^\pi(s_0)\Gamma(s_0) = \mathsf{V}(s_0) + \frac{\mathbf{Q}(s_0)\pi(s_0)\Gamma(s_0)}{1 - \gamma\boldsymbol{\eta}(s_0)\pi(s_0)}. \tag{13}$$

*where $\boldsymbol{\eta}(s_0) = \Gamma(s_0)\mathbf{P}(s_0)$, $\mathbf{Q}(s_0) = \big(\mathbf{R}(s_0) + \gamma\mathsf{V}(s_0)\mathbf{P}(s_0)\big)$. Moreover, $\boldsymbol{\eta}(s_0)\pi(s_0) \in [0, 1]$.*

*Proof:* Let $s'$ be the next state starting from $s_0$. For $\pi \in \Phi(s_0, \pi_0)$,

$$V^\pi(s_0) = \mathbb{E}\left[R(a, s_0) + \gamma\tilde{\mathsf{V}}(s'; s_0) + \gamma \cdot \gamma^{\tilde{X}(s'; s_0)} \cdot V^\pi(s_0)\right] \tag{14}$$

$$= \mathbf{R}(s_0)\pi(s_0) + \gamma\sum_{s'}\left(\mathbb{E}\big[\tilde{\mathsf{V}}(s'; s_0)\big] + \mathbb{E}\big[\gamma^{\tilde{X}(s'; s_0)}\big]V^\pi(s_0)\right)\sum_a P(s_0, a, s')\pi(a, s_0), \tag{15}$$

where $\pi(a, s_0)$ is the probability of taking action $a$ in state $s_0$ when adopting policy $\pi$ and $P(s_0, a, s')$ is the transition probability of moving to state $s'$ when taking $a$ in $s_0$.

That is,

$$V^\pi(s_0) = \mathbf{R}(s_0)\pi(s_0) + \gamma\mathsf{V}(s_0)\mathbf{P}(s_0)\pi(s_0) + \gamma V^\pi(s_0)\boldsymbol{\eta}(s_0)\pi(s_0), \tag{16}$$

where $\boldsymbol{\eta}(s_0) = \Gamma(s_0)\mathbf{P}(s_0)$, and it is easy to see $\boldsymbol{\eta}(s_0)\pi(s_0) \in [0, 1]$. Rearranging it yields

$$V^\pi(s_0) = \big(\mathbf{R}(s_0) + \gamma\mathsf{V}(s_0)\mathbf{P}(s_0)\big)\frac{\pi(s_0)}{1 - \gamma\boldsymbol{\eta}(s_0)\pi(s_0)} = \mathbf{Q}(s_0)\frac{\pi(s_0)}{1 - \gamma\boldsymbol{\eta}(s_0)\pi(s_0)}. \tag{17}$$

Note that, for all $s \in \mathcal{S}$, we have

$$V^\pi(s) = \mathbb{E}\left[\tilde{\mathsf{V}}(s; s_0) + \gamma^{\tilde{X}(s'; s_0)} \cdot V^\pi(s_0)\right] = \mathbb{E}\left[\tilde{\mathsf{V}}(s; s_0)\right] + \mathbb{E}\left[\gamma^{\tilde{X}(s'; s_0)}\right]V^\pi(s_0). \tag{18}$$

Replacing $V^\pi(s_0)$ with the expression in (17) yields (13). ∎

**Corollary 1 (Full version of Proposition 1 in the main text)** *For $\pi_0, \pi_1 \in \Phi(s_0, \pi_0)$, let $\pi_\alpha = (1 - \alpha)\pi_0 + \alpha\pi_1$, with $\alpha \in [0, 1]$. Then*

$$\mathbf{V}^{\pi_\alpha} = (1 - \beta(\alpha; s_0))\mathbf{V}^{\pi_0} + \beta(\alpha; s_0)\mathbf{V}^{\pi_1}, \tag{19}$$

*where*

$$\beta(\alpha; s_0) = \frac{\alpha\phi_1}{(1 - \alpha)\phi_0 + \alpha\phi_1} \tag{20}$$

*with $\phi_i = 1 - \gamma\boldsymbol{\eta}(s_0)\pi_i(s_0) \in [1 - \gamma, 1]$ and $\beta'(\alpha; s_0) > 0$. Besides,*

$$\mathbf{V}^{\pi_1} - \mathbf{V}^{\pi_0} = \big(V^{\pi_1}(s_0) - V^{\pi_0}(s_0)\big)\Gamma(s_0), \tag{21}$$

*and*

$$\frac{\partial\mathbf{V}^{\pi_\alpha}}{\partial\alpha} = \beta'(\alpha; s_0) \cdot (\mathbf{V}^{\pi_1} - \mathbf{V}^{\pi_0}) = \beta'(\alpha; s_0) \cdot \mathbf{D}(s_0)\Gamma(s_0) \tag{22}$$

$$= \frac{\partial V^{\pi_\alpha}(s_0)}{\partial\alpha}\Gamma(s_0), \tag{23}$$

*where*

$$\mathbf{D}(s_0) = \mathbf{Q}(s_0)\left(\frac{\pi_1(s_0)}{1 - \gamma\boldsymbol{\eta}(s_0)\pi_1(s_0)} - \frac{\pi_0(s_0)}{1 - \gamma\boldsymbol{\eta}(s_0)\pi_0(s_0)}\right). \tag{24}$$

*(Note: Eq (23) is identical to (5) in Prop 1.)*

*Proof:* Notice that

$$
(1 - \beta(\alpha; s_0))\mathbf{V}^{\pi_0} + \beta(\alpha; s_0)\mathbf{V}^{\pi_1}
$$

$$
= \quad \frac{(1-\alpha)\phi_0}{(1-\alpha)\phi_0 + \alpha\phi_1} \, \mathbf{V}^{\pi_0} + \frac{\alpha\phi_1}{(1-\alpha)\phi_0 + \alpha\phi_1} \, \mathbf{V}^{\pi_1}
$$

$$
= \quad \mathsf{V}(s_0) + \mathbf{Q}(s_0) \left[ \frac{(1-\alpha)\phi_0}{(1-\alpha)\phi_0 + \alpha\phi_1} \, \frac{\pi_0(s_0)}{\phi_0} \right.
$$

$$
\left. + \frac{\alpha\phi_0}{(1-\alpha)\phi_0 + \alpha\phi_1} \, \frac{\pi_1(s_0)}{\phi_1} \right] \Gamma(s_0) \qquad \text{[by (17)]}
$$

$$
= \quad \mathsf{V}(s_0) + \mathbf{Q}(s_0) \frac{(1-\alpha)\pi_0 + \alpha\pi_1}{(1-\alpha)\phi_0 + \alpha\phi_0} \, \Gamma(s_0)
$$

$$
= \quad \mathsf{V}(s_0) + \mathbf{Q}(s_0) \frac{\pi_\alpha}{1 - \gamma\boldsymbol{\eta}(s_0)\pi_\alpha} \, \Gamma(s_0)
$$

$$
= \quad \mathbf{V}^{\pi_\alpha} \qquad\qquad\qquad\qquad\qquad\qquad\qquad \text{[by (17)]}
$$

which is (19).

By Prop 9, $\boldsymbol{\eta}(s_0)\pi_i(s_0) \in [0, 1)$ for $i = 0, 1$. Hence, $\phi_i = 1 - \gamma\boldsymbol{\eta}(s_0)\pi_i(s_0) \in [1-\gamma, 1]$ and $\beta'(\alpha; s_0) = \frac{\phi_0\phi_1}{(\phi_0\alpha - \phi_0 - \phi_1\alpha)^2} \in [1-\gamma, 1]$. Besides, by (19), we have $\frac{\partial \mathbf{V}^{\pi_\alpha}}{\partial \alpha} = \beta'(\alpha; s_0) \cdot (\mathbf{V}^{\pi_1} - \mathbf{V}^{\pi_0})$. Plugging $\pi_1$ and $\pi_0$ into (13) followed by taking the difference yields $\mathbf{V}^{\pi_1} - \mathbf{V}^{\pi_0} = \mathbf{D}(s_0)\Gamma(s_0)$, with $\mathbf{D}(s_0)$ defined in (24). In this way, (22) is proved.

Let $\Gamma(s; s_0)$ be the entry for $s$ in $\Gamma(s_0)$. Namely, $\Gamma(s; s_0) = \mathbb{E}\big[\gamma^{\tilde{\mathsf{X}}(s; s_0)}\big]$. Then

$$
V^{\pi_1}(s_0) - V^{\pi_0}(s_0) = \mathbf{D}(s_0)\Gamma(s_0; s_0) = \mathbf{D}(s_0)\mathbb{E}\big[\gamma^0\big] = \mathbf{D}(s_0). \tag{25}
$$

Hence,

$$
\mathbf{V}^{\pi_1} - \mathbf{V}^{\pi_0} = \mathbf{D}(s_0)\Gamma(s_0) = \big(V^{\pi_1}(s_0) - V^{\pi_0}(s_0)\big)\Gamma(s_0), \tag{26}
$$

which is (21). Likewise, (22) can be written as

$$
\frac{\partial V^{\pi_\alpha}(s_0)}{\partial \alpha} = \beta'(\alpha; s_0) \cdot \mathbf{D}(s_0)\Gamma(s_0; s_0) = \beta'(\alpha; s_0) \cdot \mathbf{D}(s_0)\mathbb{E}\big[\gamma^0\big] = \beta'(\alpha; s_0) \cdot \mathbf{D}(s_0). \tag{27}
$$

Therefore, for $s \in \mathcal{S}$,

$$
\frac{\partial V^{\pi_\alpha}(s)}{\partial \alpha} = \beta'(\alpha; s_0) \cdot \mathbf{D}(s_0) \, \Gamma(s; s_0) = \frac{\partial V^{\pi_\alpha}(s_0)}{\partial \alpha}\Gamma(s; s_0), \tag{28}
$$

which is (23). ∎

**Proposition 10 (Proposition 2 in the main text)** *Assume $\pi_0 \in \Pi$ is not optimal in the associated scalar problem for all $\mathbf{w} \neq \mathbf{0}$. Let $s_0 \in \mathcal{S}$. We have the conic hull*

$$
\mathcal{Q}_1 = \text{cone}\left(V^\pi(s_0) - V^{\pi_0}(s_0)\Big|\pi \in \bigcup_{s \in \mathcal{S}} \Phi(s, \pi_0)\right) = \mathbb{R}^d, \tag{29}
$$

$$
\mathcal{Q}_2 = \text{cone}\left(V^\pi_\mu - V^{\pi_0}_\mu\Big|\pi \in \bigcup_{s \in \mathcal{S}} \Phi(s, \pi_0)\right) = \mathbb{R}^d. \tag{30}
$$

*Therefore, for all $\mathbf{u} \in \mathbb{R}^d$, we can construct a function*

$$
\pi(\alpha_1, \ldots, \alpha_M) = \left(1 - \sum_{i=1}^{M} \alpha_i\right)\pi_0 + \sum_{i=1}^{M} \alpha_i\pi_i \tag{31}
$$

*with $\alpha_i \geq 0$, $\sum_j \alpha_j \leq 1$ and $\pi_i \in \Pi$ such that*

$$
\mathbf{u} = \sum_{j=1}^{M} \frac{\partial V^{\pi(\mathbf{0})}(s_0)}{\partial \alpha_j} \cdot r_j \quad \text{for some } r_j \geq 0. \tag{32}
$$

*(Note that $\pi(\mathbf{0}) = \pi(0, 0, \ldots, 0) = \pi_0$.) Besides, assuming the MDP is ergodic, if $\gamma \to 1$,*

$$\sum_{j=1}^{M} \frac{\partial V^{\pi(\mathbf{0})}(s)}{\partial \alpha_j} \cdot r_j \to \mathbf{u} \quad \text{for all } s \in \mathcal{S}. \tag{33}$$

*Similarly, we can construct $\pi(\alpha_1, \ldots, \alpha_\mathsf{M})$ such that*

$$\mathbf{u} = \sum_{j=1}^{\mathsf{M}} \frac{\partial V_\mu^{\pi(\mathbf{0})}}{\partial \alpha_j} \cdot \mathsf{r}_j \quad \text{for some } \mathsf{r}_j \geq 0. \tag{34}$$

*Proof:* Assume the cone $\left( V^\pi(s_0) - V^{\pi_0}(s_0) \middle| \pi \in \bigcup_{s \in \mathcal{S}} \Phi(s, \pi_0) \right) \neq \mathbb{R}^d$. Then by Farkas' lemma (Dax, 1997), it must be contained in some closed half-space $\mathcal{H}_1 = \{\mathbf{x} \in \mathbb{R}^d | \mathbf{n}_1^\top \mathbf{x} \leq 0\}$. Now we show that $\mathbf{n}_1^\top V^{\pi_0}(s_0)$ cannot be further improved, which means $\pi$ is optimal in the associated scalar RL problem with weight $\mathbf{n}_1$ and thus is a contradiction. To improve $\mathbf{n}_1^\top V^\pi(s_0)$, there must be some $s' \in \mathcal{S}$ that is reachable from $s_0$ and $\pi' \in \Phi(s', \pi_0)$ such that $\mathbf{n}_1^\top (V^{\pi'}(s') - V^{\pi_0}(s')) > 0$. By (21), this implies,

$$\mathbf{n}_1^\top \left( V^{\pi'}(s_0) - V^{\pi_0}(s_0) \right) = \mathbf{n}_1^\top \left( V^{\pi'}(s') - V^{\pi_0}(s') \right) \cdot \Gamma(s_0; s') > 0, \tag{35}$$

where $\Gamma(s_0; s') > 0$ is the entry corresponding to $s_0$ in $\Gamma(s')$.[7] Hence, $V^{\pi'}(s_0) - V^{\pi_0}(s_0) \notin \mathcal{H}_1$, which is a contradiction. Therefore, we have shown that $\mathcal{Q}_1 = \mathbb{R}^d$.

Similarly, if $\mathcal{Q}_2 \neq \mathbb{R}^d$, Farkas' lemma shows it must be contained in some closed half-space $\mathcal{H}_2 = \{\mathbf{x} \in \mathbb{R}^d | \mathbf{n}_2^\top \mathbf{x} \leq 0\}$. Since $\pi_0$ is assumed to be not optimal in all associated SORL problems with $\mathbf{w} \neq \mathbf{0}$, it is not optimal for the SORL one with weight $\mathbf{n}_2$. Therefore, there exists some $s' \in \mathcal{S}$ that is reachable when following the initial distribution and $\pi' \in \Phi(s', \pi_0)$ such that $\mathbf{n}_2^\top (V^{\pi'}(s') - V^{\pi_0}(s')) > 0$. Then (21) implies

$$\mathbf{n}_2^\top (V^{\pi'}(s) - V^{\pi_0}(s)) = \mathbf{n}_2^\top (V^{\pi'}(s') - V^{\pi_0}(s')) \cdot \Gamma(s; s') > 0 \tag{36}$$

for all $s \in \mathcal{S}$. As a result,

$$\mathbf{n}_2^\top \left( V_\mu^\pi - V_\mu^{\pi_0} \right) = \mathbf{n}_2^\top \sum_{s \in \mathcal{S}} \mu(s) \left( V^{\pi'}(s) - V^{\pi_0}(s) \right) > 0. \tag{37}$$

Hence, $V_\mu^\pi - V_\mu^{\pi_0} \notin \mathcal{H}_2$, which is a contradiction. Therefore, $\mathcal{Q}_2 = \mathbb{R}^d$.

Since $\mathcal{Q}_1 = \mathbb{R}^d$, there exist $s_i \in \mathcal{S}$, $\pi_i \in \bigcup_{s \in \mathcal{S}}(s_i, \pi_0)$ and $d_i' \geq 0$, for $i = 1, \ldots, M$, such that

$$\mathbf{u} = \sum_{i=1}^{M} d_i' \left( V^{\pi_i}(s_0) - V^{\pi_0}(s_0) \right). \tag{38}$$

Define function $\pi$ taking the expression (31). By Corollary 1, we have

$$\frac{\partial \mathbf{V}^{\pi(\mathbf{0})}}{\partial \alpha_j} = \frac{\partial \mathbf{V}^{(1-\alpha_j)\pi_0 + \alpha_j \pi_j}|_{\alpha_j = 0}}{\partial \alpha_j} = \beta'(\alpha_j; s_j) \cdot (\mathbf{V}^{\pi_j} - \mathbf{V}^{\pi_0}) = \frac{\partial \mathbf{V}^{\pi(\mathbf{0})}(s_j)}{\partial \alpha_j} \Gamma(s_j). \tag{39}$$

This implies,

$$\frac{\partial V^{\pi(\mathbf{0})}(s_0)}{\partial \alpha_j} = \beta'(\alpha_j; s_j) \cdot (V^{\pi_j}(s_0) - V^{\pi_0}(s_0)) \tag{40}$$

$$= \frac{\partial V^{\pi(\mathbf{0})}(s_j)}{\partial \alpha_j} \Gamma(s_0; s_j). \tag{41}$$

Combining (38) and (40) yields

$$\mathbf{u} = \sum_{i=1}^{M} \frac{d_i'}{\beta'(\alpha_i; s_i)} \frac{\partial V^{\pi(\mathbf{0})}(s_0)}{\partial \alpha_i}. \tag{42}$$

---

[7]We note that the policy optimization is performed in $s'$ instead of $s_0$ (as we did in Cor 1).

Setting $r_i = \frac{d'_i}{\beta'(\alpha_j; s_j)}$ yields (32). Moreover, if the MDP is ergodic, as $\gamma \to 1$, we have all entries of $\Gamma(s_j)$ approach to one for all $j = 1, 2, \dots, M$. Then, combining (41) and (42) yield

$$\mathbf{u} = \sum_{i=1}^{M} \frac{d'_i}{\beta'(\alpha_i; s_i)} \frac{\partial V^{\pi(\mathbf{0})}(s_0)}{\partial \alpha_i} = \sum_{i=1}^{M} \frac{d'_i}{\beta'(\alpha_i; s_i)} \frac{\partial V^{\pi(\mathbf{0})}(s_i)}{\partial \alpha_i} = \sum_{i=1}^{M} r_i \frac{\partial V^{\pi(\mathbf{0})}(s)}{\partial \alpha_i}, \qquad (43)$$

for all $s \in \mathcal{S}$.

Likewise, since $\mathcal{Q}_2 = \mathbb{R}^d$, there exist $\mathsf{r}'_i \geq 0$ for $i = 1, \dots, \mathsf{M}$ such that

$$\mathbf{u} = \sum_{i=1}^{\mathsf{M}} \mathsf{r}'_i \left( V_\mu^{\pi_i} - V_\mu^{\pi_0} \right). \qquad (44)$$

Notice that

$$
\begin{aligned}
\frac{\partial V_\mu^{\pi(\mathbf{0})}}{\partial \alpha_j} &= \frac{\partial \sum_s \mu(s) V^{\pi(\mathbf{0})}(s)}{\partial \alpha_j} = \sum_s \mu(s) \cdot \frac{\partial V^{\pi(\mathbf{0})}(s)}{\partial \alpha_j} \\
&= \sum_s \mu(s) \beta'(\alpha_j; s_j) \cdot \left( V^{\pi_j}(s) - V^{\pi_0}(s) \right) \qquad \text{[by (22)]} \\
&= \beta'(\alpha_j; s_j) \left( V_\mu^{\pi_j} - V_\mu^{\pi_0} \right).
\end{aligned}
$$

Plugging it into (44) yields,

$$\mathbf{u} = \sum_{i=1}^{\mathsf{M}} \frac{\mathsf{r}'_i}{\beta'(\alpha_i; s_i)} \cdot \frac{\partial V_\mu^{\pi(\mathbf{0})}}{\partial \alpha_i}. \qquad (45)$$

Setting $\mathsf{r}_i = \frac{\mathsf{r}'_i}{\beta'(\alpha_i; s_i)}$ yields (34). ∎

**Proposition 11 (Proposition 3 in the main text)** *For $s \in \mathcal{S}$, $\mathbb{V}(s)$ is convex. Besides, $\mathbb{V}_\mu$ is convex.*

*Proof:* We first consider the set of policies $\tilde{\Pi}$ that are not optimal in the associated scalar problem for all $\mathbf{w} \neq \mathbf{0}$. Then for $\pi_1, \pi_2 \in \tilde{\Pi}$ and $\alpha \in [0, 1]$, we show

$$(1 - \alpha) V^{\pi_1}(s) + \alpha V^{\pi_2}(s) = V^{\pi'}(s) \quad \text{for some } \pi' \in \tilde{\Pi}. \qquad (46)$$

Note that, since $V^{\pi_1}(s)$ and $V^{\pi_2}(s)$ are not optimal in the associated SORL problem for all $\mathbf{w} \neq \mathbf{0}$, for any vector $\mathbf{v}$ on the line segment from $V^{\pi_1}(s)$ to $V^{\pi_2}(s)$, $\mathbf{v}$ cannot be optimal in any associated problem either. (Otherwise, at least one of $V^{\pi_1}(s)$ and $V^{\pi_2}(s)$ is optimal.) Therefore, starting from $V^{\pi_1}(s)$, we can keep constructing function $\tilde{\pi}$ defined in Prop 10 with $\mathbf{u} = V^{\pi_2}(s) - V^{\pi_1}(s)$ to move along the line segment.[8] In this way, we can find a policy $\pi$ that has $V^\pi(s)$ corresponding to each point on the line segment, which implies $\{V^\pi(s) | \pi \in \tilde{\Pi}\}$ is convex. Therefore, $\mathbb{V}(s) = \{V^\pi(s) | \pi \in \Pi\} = \mathrm{cl}(\{V^\pi(s) | \pi \in \tilde{\Pi}\})$ is convex.

Similarly, for $\pi_1, \pi_2 \in \tilde{\Pi}$, $V_\mu^{\pi_1}$ and $V_\mu^{\pi_2}$ are not optimal in the associated SORL problem for all $\mathbf{w} \neq \mathbf{0}$. Therefore, for any vector $\mathbf{v}$ that is a convex combination of $V_\mu^{\pi_1}$ and $V_\mu^{\pi_2}$, $\mathbf{v}$ is not optimal in any associated SORL problem either. According to Prop 10, for every convex combination $\mathbf{v}$, there is a policy $\pi$ that has $V_\mu^\pi = \mathbf{v}$. Hence, $\{V_\mu^\pi | \pi \in \tilde{\Pi}\}$ is convex, which implies its closure $\mathbb{V}_\mu$ is also convex. ∎

**Proposition 12 (Proposition 4 in the main text)** *For $s \in \mathcal{S}$, $\pi \in \Pi_s^*$ if there exists $\mathbf{w} \succ \mathbf{0}$ such that $\pi$ is optimal in the associated scalar problem. Also, if $\pi \in \Pi_s^*$, $\pi$ is optimal in an associated scalar problem with some nonzero $\mathbf{w} \succeq \mathbf{0}$. A similar statement holds for $\Pi_s^*$ replaced with $\Pi_\mu^*$.*

---

[8]A more rigorous treatment is to construct a line (path) integral with the directional derivative $\mathbf{u}$.

*Proof:* Consider the optimization problem

$$\text{maximize } \mathbf{w}^\top \mathbf{v} \quad \text{subject to } \mathbf{v} \in \mathbb{V}(s). \tag{47}$$

If $\pi$ is optimal for some associated scalar problem with weight $\mathbf{w} \succ \mathbf{0}$, then $V^\pi(s)$ is the optimal solution of problem (47) (Bertsekas, 2022, Prop 2.1.2). Since $\mathbb{V}(s)$ is convex (Prop 11), according to (Miettinen, 1998, Thm 3.1.2), $V^\pi(s)$ is Pareto optimal in $\mathbb{V}(s)$. That is, $\pi \in \Pi_s^*$.

Conversely, if $\pi \in \Pi_s^*$, then $V^\pi(s)$ is Pareto optimal in $\mathbb{V}(s)$. Since $\mathbb{V}(s)$ is convex, by (Miettinen, 1998, Thm 3.1.4), there exists $\mathbf{w} \succeq \mathbf{0}$ and $\mathbf{w} \neq \mathbf{0}$ such that $V^\pi(s)$ is a solution of problem (47). Then $\pi$ is optimal in the associated SORL problem with the nonzero $\mathbf{w} \succeq \mathbf{0}$.

Repeating the derivations by replacing $\Pi_s^*$ with $\Pi_\mu^*$ proves the similar statement for $\Pi_\mu^*$. ∎

**Proposition 13 (Proposition 5 in the main text)** *Function $g_s$ given in (9) is well-defined, surjective and uniformly continuous.*

We prove Prop 13 by first proving the functional relationship from $\mathbf{w} \in \mathbb{W}^+$ to $V_{\alpha f}^\pi(s) \in \mathbb{V}_{\alpha f}^\star(s)$ and thus $g_s$ is well defined. In particular, we show

**Lemma 1** *For any $\mathbf{w} \in \mathbb{W}^+$, there exists a unique $V_{\alpha f}^\pi(s) \in \mathbb{V}_{\alpha f}^\star(s)$ such that $\pi$ is optimal in $SORL(\mathbf{w})$. (i.e., $V_{\alpha f}^\pi(s)$ is the only element in $\mathbb{V}_{\alpha f}(s)$ having the greatest projection on $\mathbf{w}$.)*

*Proof:* We prove the lemma by contradiction. Assume that there exist two distinct $V_{\alpha f}^{\pi_1}(s)$ and $V_{\alpha f}^{\pi_1}(s)$ in $\mathbb{V}_{\alpha f}^\star(s)$ such that $\pi_1$ and $\pi_2$ are optimal in $SORL(\mathbf{w})$. Then we have

$$\{V_{\alpha f}^{\pi_1}(s), V_{\alpha f}^{\pi_2}(s)\} \subseteq \underset{\mathbf{v} \in \mathbb{V}_{\alpha f}(s)}{\arg\max} \mathbf{w}^\top \mathbf{v}, \tag{48}$$

and

$$\mathbf{w}^\top V_{\alpha f}^{\pi_1}(s) = \mathbf{w}^\top V_{\alpha f}^{\pi_2}(s). \tag{49}$$

In a SORL problem, it is well-known that a policy $\pi$ is optimal for initial state $s$ if and only if it is optimal for all states that are reachable from $s$ (Sutton & Barto, 2018). Since $V_{\alpha f}^{\pi_1}(s) \neq V_{\alpha f}^{\pi_2}(s)$, we have $\pi_1(s') \neq \pi_2(s')$ for some $s'$ that is reachable from $s$ under policies $\pi_1$ and $\pi_2$. Moreover, we have $\mathbf{w}^\top V_{\alpha f}^{\pi_1}(s') = \mathbf{w}^\top V_{\alpha f}^{\pi_2}(s')$. We then consider a new policy $\pi'$ that equals $\pi_1$ for all $s \in \mathcal{S}$ except $s'$. For $s'$, $\pi'(s') = \frac{1}{2}(\pi_1(s') + \pi_2(s'))$. Due to the strongly concavity of the immediate augmented reward, the projected immediate reward of $\pi'$ on $\mathbf{w}$ is strictly greater than the ones of $\pi_1$ and $\pi_2$. As a single-state action optimization is sufficient to increase the value function in all states (Sutton & Barto, 2018, p78), we also have $\mathbf{w}^\top V_{\alpha f}^{\pi'}(s) > \mathbf{w}^\top V_{\alpha f}^{\pi_1}(s) = \mathbf{w}^\top V_{\alpha f}^{\pi_2}(s)$, which contradicts (48). ∎

Since for each $\mathbf{w} \in \mathbb{W}^+$, there exists a unique corresponding $V_{\alpha f}^\pi(s) \in \mathbb{V}_{\alpha f}^\star(s)$. The relationship $g_s(\cdot)$ by definition is a function. We then show that

**Lemma 2** $g_s : \mathbb{W}^+ \to \mathbb{V}_{\alpha f}^\star(s)$ *is surjective.*

*Proof:* Consider a variant of the MORL problem that has the same setting as the original one except that we now treat the action-taking distributions of a state as actions. We keep referring the actions under the original definition as actions and the ones under the new definition as the d-actions.

Then for each state-action pair $(\bar{a}, s)$, the immediate reward is $\mathbf{R}(s)\bar{a} + \alpha f(\bar{a})$. This setting is the same as the one we considered in the proof of the convexity of the induced value function's range. Therefore, let $\bar{\mathbb{V}}_{\alpha f}(s)$ denote the range of the induced value function for the original MORL problem's variant; by Prop 3, we have $\bar{\mathbb{V}}_{\alpha f}(s)$ is convex. (In this variant of the MORL problem, the action space $\bar{\mathcal{A}}$ is actually infinite. Thus, to apply Prop 3, we need to first approximate the action space through sufficiently fine discretization (Chow & Tsitsiklis, 1991).) Let $\bar{\mathbb{V}}_{\alpha f}^\star(s)$ denotes the set

of PE elements in $\mathbb{V}^\star_{\alpha f}$. Then applying the same method for the proof of Prop 12, we can show that for all $\mathbf{v} \in \bar{\mathbb{V}}^\star_{\alpha f}(s)$, there exists $\mathbf{w} \in \mathbb{W}^+$ such that $\mathbf{v}$ has the maximum projection on $\mathbf{w}$. Besides, applying the same method used in the proof of Lem 1, we can show that for any $\mathbf{w} \in \mathbb{W}^+$, there exists a unique $\mathbf{v} \in \bar{\mathbb{V}}^\star_{\alpha f}(s)$ such that $\mathbf{v}$ has the maximum projection on $\mathbf{w}$. Therefore, there is a functional relationship $\bar{g}_s : \mathbb{W}^+ \to \bar{\mathbb{V}}^\star_{\alpha f}(s)$ that maps $\mathbf{w}$ to $\mathbf{v}$. Moreover, $\bar{g}_s$ is surjective. Finally, we show that $\bar{g}_s = g_s$ by showing that $\mathbf{v}$ must be in $\mathbb{V}^\star_{\alpha f}(s)$.

Given $\mathbf{v} \in \bar{\mathbb{V}}^\star_{\alpha f}(s)$ that has the maximum projection over some $\mathbf{w} \in \mathbb{W}^+$. Then consider its associated SORL with weight $\mathbf{w}$. Since the corresponding policy of taking d-actions are optimal, it must achieve the maximal cumulative discounted reward in all $s'$ that are reachable from $s$. Then suppose in some $s'$ the policy of taking actions is not deterministic (i.e., the agent follows some distribution $p(\bar{a})$ to take d-actions). Then by replacing the distribution with the one only taking d-action $\bar{a}' = \int_{\bar{A}} \bar{a} p(\bar{a})$, the projected cumulative discounted reward starting at $s'$ increases (due to the strongly convavity and the Jensen's inequality). Therefore, to find an optimal policy of the associated SORL, we only need to consider the deterministic d-action policy. Namely, it is sufficient to choose one distribution to take actions instead of adopting a bilevel design such that first follow a distribution to pick an action-taking distribution $q$ followed by using $q$ to take actions. Equivalently, we have shown that $\mathbf{v} \in \mathbb{V}^\star_{\alpha f}(s)$, which completes the proof. ∎

Finally, we show that $g_s$ is uniformly continuous.

**Lemma 3** *$g_s$ is uniformly continuous.*

*Proof:* We first show that $\mathbf{w}$ is continuous over $\mathbb{W}^+$. By Lem 1, we know $g_s(\mathbf{w})$ can be written as

$$g_s(\mathbf{w}) = \underset{\mathbf{v} \in \mathbb{V}_{\alpha f}(s)}{\operatorname{argmax}} \mathbf{w}^\top \mathbf{v}. \tag{50}$$

For a sequence $\mathbf{w}_k \to \mathbf{w}^\star$, let $\mathbf{v}_k = g_s(\mathbf{w}_k)$ and $\mathbf{v}^\star = g_s(\mathbf{w}^\star)$. Then for any subsequence $I \subset \mathbb{N}$, $\mathbf{v}_k$ has an accumulation point $\mathbf{v}'$ (because $\mathbb{V}_{\alpha f}(S)$ is closed and bounded and due to the Bolzano–Weierstrass theorem (Davidson & Donsig, 2009)). Since $\mathbf{w}_k^\top \mathbf{v}_k \geq \mathbf{w}_k^\top \mathbf{v}$ for all $\mathbf{v} \in \mathbb{V}_{\alpha f}(s)$, $k \in I$. We also have $\mathbf{w}^{\star\top} \mathbf{v}' \geq \mathbf{w}^{\star\top} \mathbf{v}$ for all $\mathbf{v} \in \mathbb{V}_{\alpha f}(s)$. Therefore, $\mathbf{v}' = \mathbf{v}^\star$ and $g_s$ is continuous by definition (Davidson & Donsig, 2009).

Finally, since $\mathbb{W}^+$ is compact, $g_s$ is uniformly continuous (Davidson & Donsig, 2009). ∎

Combining Lemma 1-3 completes the proof of Prop 13.

**Corollary 2 (Proposition 6 in the main text)** *If $\pi \in \Pi^*_s$ for some state $s \in \mathcal{S}$, then for all $s' \in \mathcal{S}$, $\pi \in \Pi^*_{s'}$. Thus, the sets of SPE policies coincide for all initial states and is a subset of $\Pi^*$.*

*Proof:* According to Prop 12, if $\pi \in \Pi^*_{s'}$ for some $s' \in \mathcal{S}$, then $V^\pi(s')$ is optimal in an associated SORL with some none-zero $\mathbf{w} \succeq \mathbf{0}$. Thus, $V^\pi(s)$ is the optimal solution of problem (47) for all $s \in \mathcal{S}$ (by (Bertsekas, 2022, Prop 2.1.2) and the ergodic assumption).

If $\mathbf{w} \succ \mathbf{0}$, then by Prop 12, $V^\pi(s) \in \mathbb{V}(s)$ for all $s \in \mathcal{S}$. That is, $\pi \in \Pi^*_s$ for all $s$.

If $\mathbf{w}$ contains zero entries, as $\pi(s') \in \Pi^*_{s'}$, $V^\pi(s')$ is Pareto optimal in $\mathbb{V}(s')$. Intuitively, $V^\pi(s')$ can be seen as a maximizer of problem (47) with weight $\mathbf{w}' = \lim_{\epsilon \to +0} \mathbf{w} + \epsilon \mathbf{1}$. Then since $\mathbf{w} + \epsilon \mathbf{1} \succ \mathbf{0}$ as it approaches to $\mathbf{w}$, the problem is reduced to the first case, and thus, $\pi \in \Pi^*_s$ for all $s$.

A more rigorous proof can be done by introducing a lexicographic order over $\mathbb{R}^d$. Specifically, we need to construct a lexicographic order such that $V^\pi(s')$ is the global maximum $\mathbb{V}(s') \subset \mathbb{R}^d$. Let $\mathbf{w}_0 = \mathbf{w}$, $\mathsf{S}_0 = V^\pi(s')$ and $\mathsf{m}_0 = \max_{\mathbf{v} \in \mathsf{S}_0} \mathbf{w}_0^\top \mathbf{v}$. Define $\mathsf{S}_1 = \{\mathbf{v} \in \mathcal{S}_0 | \mathbf{w}_0^\top \mathbf{v} = \mathsf{m}_0\}$. Since $\mathsf{S}_0$ is convex, so is $\mathsf{S}_1$ as it is an intersection of $\mathsf{S}_0$ and a hyperplane. It is easy to check $V^\pi(s') \in \mathsf{S}_1$ and is Pareto optimal in $\mathsf{S}_1$. Applying (Miettinen, 1998, Thm 3.1.4), there exists nonzero $\mathbf{w}_1 \succeq \mathbf{0}$ such that $\mathbf{w}_1 \cdot \mathbf{w}_0 = 0$ and $\mathbb{V}(s')$ is a solution of

$$\text{maximize } \mathbf{w}_1^\top \mathbf{v} \quad \text{subject to } \mathbf{v} \in \mathsf{S}_1. \tag{51}$$

We can continue this process to get $\mathsf{m}_k = \max_{\mathbf{v} \in \mathsf{S}_k} \mathbf{w}_k^\top \mathbf{v}$, $\mathsf{S}_{k+1} = \{\mathbf{v} \in \mathsf{S}_k | \mathbf{w}_k^\top \mathbf{v} = \mathsf{m}_k\}$, followed by applying (Miettinen, 1998, Thm 3.1.4) to get $\mathbf{w}_{k+1}$, where $\mathbf{w}_{k+1} \cdot \mathbf{w}_j = 0$ for all $j < k+1$. Besides, $\mathbf{w}_{k+1}$ has at least one nonzero entry, which is zero for the counterpart in $\mathbf{w}_j$ for all $j < k+1$.

The process will continue until we get a list of non-zero weights $\mathbf{w}_0, \mathbf{w}_1, \ldots, \mathbf{w}_n \succeq \mathbf{0}$ such that for any dimension $i = 1, 2, \ldots, d$, there is exactly one $\mathbf{w}_j$ having a positive entry in dimension $i$. Then we can construct a lexicographic order $\preceq_L$ over $\mathbb{R}^d$ by ordering tuple $(\mathbf{w}_0^\top \mathbf{v}, \mathbf{w}_1^\top \mathbf{v}, \ldots, \mathbf{w}_n^\top \mathbf{v})$ where $\mathbf{v} \prec_L \mathbf{u}$ if $\mathbf{w}_k^\top \mathbf{v} = \mathbf{w}_k^\top \mathbf{v}$ for $k < j$ and $\mathbf{w}_j^\top \mathbf{v} < \mathbf{w}_j^\top \mathbf{v}$.

Since $V^\pi(s')$ is Pareto optimal in all $\mathsf{S}_k$, it is the global maximum of $\mathbb{V}(s')$ under $\preceq_L$. Since for any dimension in $\mathbb{R}^d$, there is exactly one $\mathbf{w}_j$, $j = 0, 1, \ldots, n$, having a positive entry. For $\mathbf{u}, \mathbf{v} \in \mathbb{R}^d$, if $\mathbf{u} \neq \mathbf{u}$, either $\mathbf{u} \succ_L \mathbf{v}$ or $\mathbf{u} \prec_L \mathbf{v}$. Thus, $\mathbb{V}(s')$ is in fact the unique global maximum of $\mathbb{V}(s')$.

Gabor et al. (Gábor et al., 1998) extends the SORL setting to MORL by ordering $\mathbb{R}^d$ using a lexicographic order. They generalized the Bellman optimality operator by choosing the action that gives the greatest (vectored) value function under the lexicographic order. They showed that under this setting, the value function converges to the unique optimal point (under the lexicographic order) and the generalized Bellman optimality operator is monotonic. Thus, the value function reaches the optimality for all states simultaneously under the ergodic assumption.

As a result, since $V^\pi(s')$ is optimal in $\mathbb{V}(s')$, $V^\pi(s)$ is also optimal in $\mathbb{V}(s)$ for all $s \in \mathcal{S}$ (under order $\preceq_L$). Thus, $V^\pi(s)$ is Pareto optimal in $\mathbb{V}(s)$. (Otherwise, if there is some $\mathbf{u} \in \mathbb{V}(s)$ such that $\mathbf{u} \succ V^\pi(s)$, we can show that $\mathbf{u} \succ_L V^\pi(s)$ as well.) ∎

**Proposition 14 (Proposition 7 in the main text)** *For all $s \in \mathcal{S}$, $\Pi_s^* = \Pi_\mu^*$.*

*Proof:* If $\pi \in \Pi_{s'}^*$ for some $s' \in \mathcal{S}$, then by Prop 12, $\pi$ is optimal in an associated SORL problem with some nonzero $\mathbf{w} \succeq \mathbf{0}$. Therefore, $\mathbf{w}^\top V^\pi(s)$ reaches the maximum for all $s \in \mathcal{S}$; thus, so is $\mathbf{w}^\top V_\mu^\pi = \sum_{s \in \mathcal{S}} \mu_s \mathbf{w}^\top V^\pi(s)$.

If $\mathbf{w} \succ \mathbf{0}$, by Prop 12, $V_\mu^\pi$ is Pareto optimal in $\mathbb{V}_\mu$. Otherwise, $\mathbf{w}$ contains zero entries. Intuitively, we can approximate $\mathbf{w}$ with $\mathbf{w}' = \lim_{\epsilon \to +0} \mathbf{w} + \epsilon \mathbf{1}$ like what we do in the proof of Cor 2. More rigorously, we can use the same trick to construct a lexicographic order $\preceq_L$ over $\mathbb{R}^d$, where $V^\pi(s')$ is the only global maximum of $\mathbb{V}(s')$. Then, by Gabor et al. (Gábor et al., 1998)'s work, $V^\pi(s)$ is the only global maximum of $\mathbb{V}(s)$ for all $s$, which implies $V_\mu^\pi$ is the only global maximum in $\mathbb{V}_\mu$ under order $\preceq_L$. Therefore, $V_\mu^\pi$ is Pareto optimal in $\mathbb{V}_\mu$.

Reversely, given $\pi \in \Pi_\mu^*$, we have $V_\mu^\pi$ is Pareto optimal in $\mathbb{V}_\mu$. Then Prop 12 shows $\pi$ is optimal in an associated SORL problem with a nonzero $\mathbf{w} \succeq \mathbf{0}$. If $\mathbf{w} \succ \mathbf{0}$, then by Prop 12, $\pi \in \Pi_s^*$ for all $s \in \mathcal{S}$. Otherwise, we construct a lexicographic order $\preceq_L$ through the method used in Cor 2's proof. Then we can show that $V_\mu^\pi$ is the the global maximum in $\mathbb{V}_\mu$, and $V^\pi(s)$ is the global maximum in $\mathbb{V}(s)$ for all $s$ (under order $\preceq_L$). Thus, $V^\pi(s)$ is Pareto optimal in $\mathbb{V}(s)$ and $\pi \in \Pi^*(s)$ for all $s$. ∎

**Proposition 15 (Proposition 8 in the main text)** *For any $s \in \mathcal{S}$, as $\gamma \to 1$, the SPE set $\Pi_s^*$ approaches to the APE one $\Pi^*$.*

*Proof:* It is sufficient to show as $\gamma \to 1$, every policy $\pi \in \Pi^*$ must be in $\Pi_s^*$. In particular, assume there is $\pi \in \Pi^*$ but $\pi \notin \Pi_s^*$. Then there exists $\mathbf{v} \in \mathbb{V}(s)$ such that $\mathbf{v} \neq V^\pi(s)$ and $\mathbf{v} \succeq V^\pi(s)$. By Prop 11, $\mathbb{V}(s)$ is convex. Thus, the line segment with the ends $\mathbf{v}$ and $V^\pi(s)$ is contained in $\mathbb{V}(s)$. By Prop 10, we can optimize the policy to move from $V^\pi(s)$ to $\mathbf{v}$ over the line segment; at the same time, as $\gamma \to 1$, $V^\pi(s')$ will move in the same direction for all $s' \in \mathcal{S}$ (see (33)). Hence, the value function are improved in all states, which implies $\pi$ cannot be APE. ∎

## C THE RELATIONSHIP BETWEEN THE CONVEXITY OF THE OCCUPATION MEASURE AND THE INDUCED FUNCTION'S RANGE

The convexity of the induced function's range can also be seen as a corollary of the occupation measure's convexity, which was initially proved by Kallenberg (1983). In this section, we give a second proof of Prop 3 based on the occupation measure's convexity.

For any initial distribution $\mu$, the occupation measure of a policy $\pi$ is defined as (Altman, 1999, p27)

$$x(\pi, \mu; a, s) = (1 - \gamma) \sum_{t=0}^{\infty} \gamma^t P(a_t, s_t; \pi, \mu), \tag{52}$$

where $P(a_t, s_t; \pi, \mu)$ denotes the probability of taking action $a_t$ in state $s_t$ at step $t$ when adopting policy $\pi$ with initial distribution $\mu$. It is easy to see that we can write

$$V_\mu^\pi = \frac{1}{1 - \gamma} \sum_{s \in \mathcal{S}} \sum_{a \in \mathcal{A}} x(\pi, \mu; a, s) \cdot R(a, s). \tag{53}$$

Let $X(\pi, \mu) \in \mathbb{R}^{|\mathcal{A}| \times |\mathcal{S}|}$ such that $[X(\pi, \mu)]_{as} = x(\pi, \mu; a, s)$. Additionally, let $\mathbf{X}(\mu) = \{X(\pi, \mu) | \pi \in \Pi\}$. Then Kallenberg (1983) proved that (Altman, 1999, Thm 3.2)

**Lemma 4 (Convexity of occupation measure)** $\mathbf{X}(\mu)$ *is convex.*

In other words, for $X_1, X_2 \in \mathbf{X}(\mu)$, we also have $\delta X_1 + (1 - \delta) X_2 \in \mathbf{X}(\mu)$ for all $\delta \in [0, 1]$.

Then, we use this result to prove the convexity of the induced function's range.

*Proof:* [The second proof of Prop 3] Given $V_\mu^{\pi_1}, V_\mu^{\pi_2} \in \mathbb{V}_\mu$, by (53), we have

$$V_\mu^{\pi_1} = \frac{1}{1 - \gamma} \sum_{s \in \mathcal{S}} \sum_{a \in \mathcal{A}} x(\pi_1, \mu; a, s) \cdot R(a, s), \tag{54}$$

$$V_\mu^{\pi_2} = \frac{1}{1 - \gamma} \sum_{s \in \mathcal{S}} \sum_{a \in \mathcal{A}} x(\pi_2, \mu; a, s) \cdot R(a, s). \tag{55}$$

Then for any $\delta \in [0, 1]$, we have

$$(1 - \delta)V_\mu^{\pi_1} + \delta V_\mu^{\pi_2} = \frac{1}{1 - \gamma} \sum_{s \in \mathcal{S}} \sum_{a \in \mathcal{A}} \left[ (1 - \delta)\, x(\pi_1, \mu; a, s) + \delta\, x(\pi_2, \mu; a, s) \right] \cdot R(a, s). \tag{56}$$

Due to the convexity of $\mathbf{X}(\mu)$, we know that there exists some $\pi'$ such that

$$x(\pi', \mu; a, s) = (1 - \delta)\, x(\pi_1, \mu; a, s) + \delta\, x(\pi_2, \mu; a, s) \tag{57}$$

for all $s \in \mathcal{S}$ and $a \in \mathcal{A}$. That is,

$$(1 - \delta)V_\mu^{\pi_1} + \delta V_\mu^{\pi_2} = \frac{1}{1 - \gamma} \sum_{s \in \mathcal{S}} \sum_{a \in \mathcal{A}} x(\pi', \mu; a, s) \cdot R(a, s) = V_\mu^{\pi'} \in \mathbb{V}_\mu, \tag{58}$$

as well. Therefore, $\mathbb{V}_\mu$ is convex. Let $\mu$ be the one-hot distribution that puts all mass on state $s$. We have $\mathbb{V}(s) = \mathbb{V}_\mu$, which is also convex. ∎

## D WHY A DISCONTINUOUS $g_s$ MAKES THE TRAINING PROCESS UNSTABLE

Training using LS is unstable because of the sensitivity of $g_s$ to preference vectors $\mathbf{w}$ that are near normal to a surface or edge of $\mathbb{V}(s)$: slight changes in $\mathbf{w}$ means the optimal policy will be on a different vertex of the polytope. This behaviour will be exacerbated by estimating $\mathbb{V}(s)$ since slight changes in the value functions will also change the surface geometry. More specifically, $g_s(\mathbf{w})$ is discontinuous when we do not add our concave regularization term ($\alpha = 0$), so it is highly sensitive to errors in our estimate of $\mathbb{V}_{\alpha f}(s)$.

# E  THE SELECTION OF THE STRONGLY CONCAVE TERMS IN CAPQL

In this section, we discuss our selection of the strongly concave term used in CAPQL.

According to the theoretical results presented in Sec 4.1, $f$ can be any strongly concave term for the definition of $V^\pi$ given in (8), and Prop 5 shows that the corresponding $g_s$ defined in (9) is surjective and uniformly continuous. In this way, all the problems of the existing LS-based methods discussed in Sec 4.1 are solved.

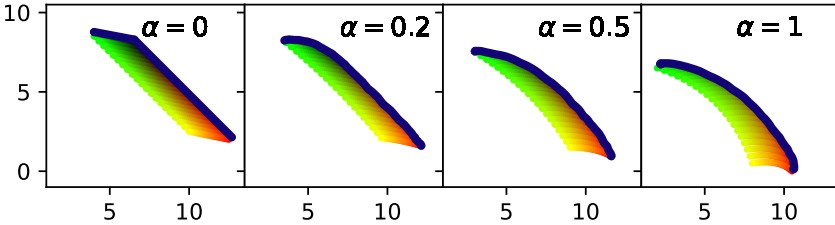

Figure 10: The effects on the induced value functions at $s_0$ for selected policies in Ex 1 by adding strongly concave terms to the immediate rewards with different alpha. Here, $f : \Delta^{|\mathcal{A}|} \to \mathbb{R}$ is defined as $f(\mathbf{p}) = -\sum_{i=1}^{|\mathcal{A}|} p_i^2$. $\mathbb{V}_{\alpha f}^\star(s_0)$ is marked in blue and the dots of the same colour among the four plots correspond to the same policy.

In Fig 10, we visualize the changes of $\mathbb{V}_{\alpha f}^\star(s_0)$ by choosing a different $f(\mathbf{p}) = -\sum_{i=1}^{|\mathcal{A}|} p_i^2$. We can observe that the transformation of $\mathbb{V}_{\alpha f}^\star(s_0)$ here is very similar to the one presented in Fig 7 where $f(\mathbf{p}) = \mathcal{H}(\mathbf{p})$.

While the theoretical results presented in Sec 4.1 holds as long as $f$ is strongly concave, some special choice of $f$ can significantly simplify the implementation of CAPQL and improve the computational efficiency. Specifically, CAPQL consists of two major parts: 1) optimizing the generalized Q-nework using the Bellman operation and 2) optimizing the policy networks $\pi_\phi$ conditioned on $\mathbf{w}$ by minimizing the KL divergence from the predicted action-taking distribution $\pi_\phi(s, \mathbf{w})$ and the target one induced by the Q-values:

$$\pi^*(s) = \operatorname*{argmax}_{\pi(s)} \left( \mathbf{w}^\top \sum_{a \in \mathcal{A}} Q_\theta(s, a, \mathbf{w}) \cdot \pi(a, s) \right) + \alpha f\big(\pi(s)\big), \tag{59}$$

where $\pi(a, s)$ denotes the probability of taking action $a$ in state $s$.[9]

Regarding the first part, the optimization of $Q(s_t, a_t, \mathbf{w})$ needs the estimated value function conditioned on $\mathbf{w}$ at the next state $s_{t+1}$. That is,

$$V(s_{t+1}, \mathbf{w}) = \mathbb{E}_{a \sim \pi_\phi(s_{t+1}, \mathbf{w})} \Big[ Q_\theta(s_{t+1}, a, \mathbf{w}) \Big] + \alpha f\Big( \pi_\phi(s_{t+1}, \mathbf{w}) \Big), \tag{60}$$

which requires $f(\pi_\phi(s_{t+1}, \mathbf{w}))$ can be computed efficiently. If the number of actions is huge or infinite, we need to estimate $V(s_{t+1}, \mathbf{w})$ through sampling. Note that, when $f = \mathcal{H}$, we have

$$V(s_{t+1}, \mathbf{w}) = \mathbb{E}_{a \sim \pi_\phi(s_{t+1}, \mathbf{w})} \Big[ Q_\theta(s_{t+1}, a, \mathbf{w}) \Big] - \alpha \, \mathbb{E}_{a \sim \pi_\phi(s_{t+1}, \mathbf{w})} \Big[ \log \, \pi_\phi(s_{t+1}, \mathbf{w}) \Big], \tag{61}$$

$$= \mathbb{E}_{a \sim \pi_\phi(s_{t+1}, \mathbf{w})} \Big[ Q_\theta(s_{t+1}, a, \mathbf{w}) - \alpha \log \pi_\phi(s_{t+1}, \mathbf{w}) \Big] \tag{62}$$

$$\approx Q_\theta(s_{t+1}, a, \mathbf{w}) - \alpha \log \pi_\phi(s_{t+1}, \mathbf{w}) \tag{63}$$

with $a \sim \pi_\phi(s_{t+1}, \mathbf{w})$. (We adopt this approximation in our CAPQL implementation as the MoJuCo environment has a continuous action space.) However, a similar estimation cannot be made given $f(\mathbf{p}) = -\sum_{i=1}^{|\mathcal{A}|} p_i^2$ because the expression is not an expectation over the action-taking distribution.

For the second part, a practical implementation requires that the $\pi^*(s)$ defined in (59) can be evaluated efficiently. When $f = \mathcal{H}$, the $\pi^*$ has the expression:

$$\pi^*(a, s) = \frac{\exp \big( \mathbf{w}^\top Q_\theta(s, a, \mathbf{w}) / \alpha \big)}{Z}, \tag{64}$$

---

[9]See Appx F for the implementation details of $\pi_\phi$ and $Q_\theta$.

where $Z = \int_{a \in \mathcal{A}} \exp\left(\mathbf{w}^\top Q_\theta(s, a, \mathbf{w})/\alpha\right) \mathrm{d}a$. In Appx F, we will show that $Z$ is not required to be evaluated explicitly. As a result, $\pi^*$ can be computed effortlessly.

Since setting $f = \mathcal{H}$ enables the efficient computation/estimation in training the Q network and the policy network, we use the entropy function to augment the immediate rewards in our CAPQL implementation.

## F    IMPLEMENTATION DETAILS OF CAPQL

In this section, we provide extra implementation details of the CAPQL algorithm and gives its psuedocode.

Our algorithm largely follows the spirit of the implementation of the soft actor critic (SAC) (Haarnoja et al., 2018). The major difference is that in CAPQL, the Q-network and the policy network are conditioned on the preference weight $\mathbf{w}$. As a result, the Q-network takes input $(s, a, \mathbf{w})$ instead of $(s, a)$, and the policy network has input $(s, \mathbf{w})$ instead of $s$.

Let $Q_\theta$ denote the Q-network with parameter $\theta$ for training, $Q_{\bar\theta}$ target network with parameter $\bar\theta$ and $\pi_\psi$ the policy network with parameter $\psi$. Additionally, let $D_\phi$ be a weight sampling distribution with support $\phi$. As mentioned in the main text, without the loss of generality, we assume that for all $\mathbf{w} \in \phi$, we have $\|\mathbf{w}\|_1 = 1$; in practice, the assumption can be simply satisfied by normalize $\mathbf{w}$ after sampling it. Then we give the CAPQL's implementation in Alg 1.

---

**Algorithm 1:** The CAPQL implementation

---

**Input:** A weight sampling distribution $D_\phi$
Initialize parameter vectors $\theta_1, \theta_2, \bar\theta_1, \bar\theta_2, \psi$
$\bar\theta_i \leftarrow \theta_i$   for $i \in \{1, 2\}$
**for** *each iteration* **do**
    **for** *each environment step* **do**
        sample $\mathbf{w} \sim D_\phi$
        $a_t \sim \pi_\psi(s_t, \mathbf{w})$
        $s_{t+1} \sim P(a_t, s_t)$
        $\mathcal{D} \leftarrow \mathcal{D} \cup \{(s_t, a_t, R(a_t, s_t), s_{t+1}, \mathbf{w})\}$
    **end**
    **for** *each training step* **do**
        $\mathcal{S} \leftarrow$ sample $N$ transitions from $\mathcal{D}$
        $\theta_i \leftarrow \theta_i - \lambda_Q \nabla_{\theta_i}\left(\frac{1}{2}\mathbb{E}_{\mathcal{S}}\|\hat{Q}(s_j, a_j, \mathbf{w}) - Q_{\theta_i}(s_j, a_j, \mathbf{w})\|_2^2\right)$   for $i \in \{1, 2\}$
            where $\hat{Q}(s_j, a_j, \mathbf{w}) = R(a_j, s_j) +$
                $\gamma\left(\min_{i \in \{1,2\}} Q_{\bar\theta_i}(s_{j+1}, a_{j+1}, \mathbf{w}) - \alpha \log \pi_\phi(a_{j+1}, s_{j+1}, \mathbf{w})\,\mathbf{1}\right)$
            and $a_{j+1} \sim \pi_\psi(s_{j+1}, \mathbf{w})$
        $\psi \leftarrow \psi - \lambda_\pi \nabla_\psi \mathbb{E}_{\mathcal{S}}\left(D_{\mathrm{KL}}\left(\pi_\psi(\cdot, s_j, \mathbf{w})\,\big\|\,\frac{\exp(\mathbf{w}^\top \min_{i \in \{1,2\}} Q_{\theta_i}(s_j, \cdot, \mathbf{w})/\alpha)}{Z(s_j, \mathbf{w})}\right)\right)$
            with $Z(s_j, \mathbf{w}) = \int_{\mathcal{A}} \exp\left(\mathbf{w}^\top \min_{i \in \{1,2\}} Q_{\theta_i}(s_j, a, \mathbf{w})/\alpha\right)\,\mathrm{d}a$
        $\bar\theta_i \leftarrow \tau\theta_i + (1-\tau)\bar\theta_i$   for $i \in \{1, 2\}$
    **end**
**end**

---

Note that when computing the gradient of $\psi$, the partition function $Z(s_j, \mathbf{w})$ does not depend on $\psi$ and will be dropped in the actual implementation. Besides, we use an exponentially moving average with a smoothing constant $\tau$ to update the target network, which stabilizes the learning trajectory. This technique has been commonly adopted in the prior work (Mnih et al., 2015; Lillicrap et al., 2016; Haarnoja et al., 2018).

**Implementation of the Q-network.**    Inspired by the design of SAC, our Q-network consists of two fully connected networks (FCNs). The two networks have the same architecture but different parameters. Each of them has two hidden layers and takes input of the dimension equal to the sum

of the ones of observation states and reward. The output dimension equals the reward's. When performing inference, the Q-network returns element-wise minimum over the two FCNs' outputs.

**Implementation of the policy network.** The policy network is implemented using the reparameterization trick. In particular, it can be written as

$$a_t = f_\psi(\epsilon_t; s_t, \mathbf{w}), \tag{65}$$

where $\epsilon_t$ is a sample of a spherical Gaussian distribution. In our implementation, we first use the trick to generate a Gaussian sample with mean $\mu_\phi(s_t, \mathbf{w})$ and standard deviation $\sigma_\phi(s_t, \mathbf{w})$ ($\mu_\phi$ and $\sigma_\phi$ are respectively implemented by a two-layer FCN), followed by using $\tanh$ function to ensure $a_t$ is in $[-1, 1]^d$ as required by the Mujoco environment.

## G  THE CONVERGENCE PROPERTIES OF CAPQL

In this section, we discuss the convergence property of the CAPQL algorithm. As mentioned in Sec 5.2, the learning task of $\text{SORL}(\mathbf{w})$ specified in (10) is the one used in SAC (Haarnoja et al., 2018); therefore, after picking $\mathbf{w}$, we use the SAC method to train the policy and the Q-network.

Let $\mathcal{W} = \{\mathbf{w}'|\mathbf{w}' \in \mathbf{\Phi} \text{ and } \|\mathbf{w}'\|_1 = 1\}$. It has been shown that by repeatedly applying the SAC optimization step, the policy converges to the optimal (Haarnoja et al., 2018, Thm 1). As a result, for CAPQL, given a fixed weight $\mathbf{w} \in \mathcal{W}$, we have the conditioned policy $\pi(\cdot, \cdot, \mathbf{w})$ converge to the optimal policy $\pi_{\mathbf{w}}^*$ such that $\mathbf{w}^\top Q^{\pi_{\mathbf{w}}^*}(s, a) \geq \mathbf{w}^\top Q^\pi(s, a)$ for all $\pi \in \Pi$ and $(s, a) \in \mathcal{S} \times \mathcal{A}$.

As the CAPQL algorithm converges conditioned on every $\mathbf{w} \in \mathcal{W}$, it also converges as a whole.

## H  ADDITIONAL EMPIRICAL STUDY OF CAPQL

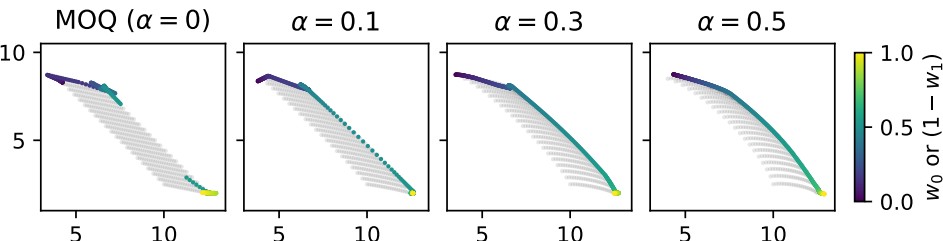

Figure 11: Visualization of the estimated $g_{s_0}(\mathbf{w})$ in Ex 1 by MOQ (Abels et al., 2019) and CAPQL with $\alpha = 0.1, 0.3, 0.5$. The MOQ can be seen as a special case of CAPQL when $\alpha \to 0$.

In this section, we provide extra empirical evidence to corroborate our theoretical results discussed in Prop 5 and Rmk 4 and show that CAPQL is capable of finding all the PE policies

We perform the experiments on the environment specified in Ex 1 which has a simple configuration so that we can plot the range of its induced value function.

In Fig 11, we visualize the estimated $g_{s_0}(\mathbf{w})$ of the CAPQL with $\alpha = 0.1, 0.3, 0.5$. We also provide one for MOQ (Abels et al., 2019) which can be seen as a limiting case of CAPQL when $\alpha \to 0$. The visualization was made by picking $\mathbf{w} \in \{[w_0, w_1]^\top \in \mathbb{R}^2 | w_0 + w_1 = 1, w_0 \in [0, 1]\}$.

Fig 11 (left plot) shows that the MOQ algorithm cannot precisely estimate $g_{s_0}$ as it is not continuous when $\alpha = 0$. In particular, the target $g_{s_0}$ has the range only consisting of the upper two vertices and the one on the right (also see Fig 6). However, the estimated $g_{s_0}$ randomly fluctuates at the vertices and mistakenly has a continuous transition between the upper two. The fluctuations suggest that the estimated $g_{s_0}$ is numerically unstable and could harm the performance of the induced policy. Additionally, the continuous transition indicates that the target $g_{s_0}$ is inaccurately estimated.

The right three plots show that the aforementioned problems can be alleviated by the CALQL algorithm. We observe that even a relatively small alpha ($\alpha = 0.1$) can largely alleviate the numerical

stability problem as the most of fluctuations near the vertices disappeared and the estimation of $g_{s_0}$ improves. We also observe that as $\alpha$ increases to $0.5$, a nearly perfect estimation of $g_{s_0}$ is obtained. The observations tell us by adding the entropy term to the immediate reward can indeed make it easier to learn $g_{s_0}$ and improves the numerical stability, which supports our claims in Prop 5 and Rmk 4.

Moreover, we can observe that the learned $g_{s_0}$ is a surjective function from $\mathbb{W}^+$ to the entire Pareto front, which implies that the corresponding policies learned by CAPQL are the induced value functions which cover the entire Pareto front of $\mathbb{V}^\star_{\alpha f}(s_0)$ as well.

## I   HYPERPARAMETERS

Tables 1-3 list the hypermeters of the models considered in Sec 5.3. Table 4 lists the specifications of the four Mujoco environments.

Table 1: Hyperparameters of CAPQL and QEnv-ctn

| Parameter | Value |
|---|---|
| Optimizer | Adam |
| learning rate | $3 \times 10^{-4}$ |
| discount factor ($\gamma$) | 0.99 |
| hidden dim (for all networks) | 256 |
| replay buffer size | $10^6$ |
| minibatch size | 256 |
| nonlinearity | ReLU |
| target smoothing coefficient ($\tau$) | 0.005 |

Table 2: Augmentation strength of CAPQL

| Environment | $\alpha$ |
|---|---|
| Hopper | 0.2 |
| Walker2D | 0.05 |
| HalfCheetah | 0.1 |
| Humanoid | 0.005 |

**Package Versioning**   Python 3.10.4 was used as the primary programming language. We accessed MuJoCo210 through gym-0.21.0's wrapper classes. Training was done using pytorch-1.12.1 and NVIDIA's CUDA 11.6.

Table 3: Hyperparameters of QEnv and MOQ

| Parameter | Value | Additional Info |
|---|---|---|
| Optimizer | Adam — SGD | SGD was only used on Humanoid because Adam was requiring too much RAM for the system |
| learning rate | $1 \times 10^{-4}$ | |
| discount factor ($\gamma$) | 0.99 | |
| hidden dim (for all networks) | A linear function of $\dim(\mathcal{S}) + d$ | The NN architecture was used in the original QEnv work. |
| replay buffer size | 4000 | |
| minibatch size | $16 \times 16$ | Effective batch size after taking the Cartesian product with the preference vectors |
| nonlinearity | ReLU | |
| # discretized values for each action dim. | 5 (2 for humanoid) | The Humanoid's action-space has dimension of 17, making it infeasible to use a more fine-grained discretization of its action space. |

Table 4: Specifications of Environments

| Environment | Action dim. | Obs. state dim. | Rwd. dim. | Rwd entries |
|---|---|---|---|---|
| Hopper | 3 | 11 | 3 | forward reward
negative ctrl cost
healthy reward |
| Walker2D | 6 | 17 | 3 | forward reward
negative ctrl cost
healthy reward |
| HalfCheetah | 6 | 17 | 2 | forward reward
negative ctrl cost |
| Humanoid | 17 | 376 | 4 | forward reward
negative ctrl cost
healthy reward
negative contact cost |

