# OpenReview forum: "Multi-Objective Reinforcement Learning: Convexity, Stationarity and Pareto Optimality"
_ICLR.cc/2023/Conference — ICLR 2023 poster_

### Official Review · Reviewer_gKVf · 2022-10-24

**Confidence:** 3
**Correctness:** 3
**Technical Novelty And Significance:** 4
**Empirical Novelty And Significance:** 2
**Recommendation:** 8

**Clarity, Quality, Novelty And Reproducibility:**

The paper is very pleasant to read and has high clarity and quality. Using a running toy example to provide intuition for the theoretical results is very helpful.

The theoretical results seem novel and significant, although the algorithm is not very novel.

Typo:
- at the bottom of page 6, g_s is not defined.



**Strength And Weaknesses:**

I am not an expert in MORL. But after reading this paper, I envision that the contributions of this paper will have a large impact on the MORL community. They can accelerate the research in this field. Furthermore, as many practical RL problems can be better formulated using multiple objectives, the contributions are very important to the RL community in general.

The authors offer several novel theoretical results, including

- The induced value function’s range for stationary policies is concave. This has been thought hard to characterize and of irregular shapes.
- There are at least three existing definitions of Pareto optimal policies. The authors rigorously distinguish and consolidate them.

I really like the new theoretical results. However, I have some questions regarding the Problems of the existing LS-based algorithms paragraph and the proposed algorithm.

- The solutions to linear programs are almost always on the vertices. If this does not make solving them less numerically stable, why would LS-based algorithms be less numerically stable?
- The motivation of identifying all SPE policies is not very clear to me. When is this required in practice?
- Maybe derived from a different perspective, based on Eq (9), the proposed algorithm CAPQL is almost the same as entropy-regulated RL algorithms. Is this correct?


**Summary Of The Paper:**

This paper conducts a rigorous analysis on the fundamental properties of the policy induced value functions in multi-objective reinforcement learning (MORL). Based on the analysis, the authors further distinguish and consolidate three existing definitions of Pareto optimal policies and identify issues of training policies via linear scalarization (LS). The authors fix this issue by adding strongly concave terms to the immediate rewards and verify the effectiveness of the approach via experiments on simulated control tasks.

**Summary Of The Review:**

Due to the strong theoretical contributions, I recommend acceptance.

---

> ### Author Response · Authors · 2022-11-12
> **Clarifications**
>
> Thank you very much for your comments and the time you have taken to review our work. We are happy to know that our work gives an intuitive grounding for some of the less obvious results and that you find our theory interesting and impactful. We have carefully gone through your comments and made the following changes and responses.
>
> **Q1:** *The solutions to linear programs are almost always on the vertices. If this does not make solving them less numerically stable, why would LS-based algorithms be less numerically stable?*
>
> **A1:** The instability occurs for weighting vectors ($\mathbf{w}$) that are near normal to a surface/edge of $\mathbb{V}(s)$: slight changes in $\mathbf{w}$ means the optimal policy will be on a different vertice of the polytope. This behaviour will be exacerbated by estimating $\mathbb{V}(s)$ since slight changes in the value functions will also change the surface geometry. More specifically, $g_s(\mathbf{w})$ is discontinuous when we do not add our concave regularization term ($\alpha = 0$), so it is highly sensitive to errors in our estimate of $\mathbb{V}_{\alpha f}(s)$.
>
>
> In the revised version, we have added our clarification here in Appx D and visualized this phenomenon in Figure 11 Appx H.
>
>
> **Q2:** *The motivation of identifying all SPE policies is not very clear to me. When is this required in practice?*
>
> **A2:** Our true motivation is to provide a method that can train the policy network to match any point on the Pareto front of the induced function's range. We want to do so because it has been reported that the existing LS-based algorithm cannot produce desired policies [1]. Our work suggests that this is because the vanilla LS-based method can only find the PE policy with the value function located on the vertexes, which is usually not desired in practice. For example, suppose we are working on a robotics problem that has objectives for battery life and performance. The policies on the vertices may only focus on maximizing the battery life or performance, which is usually not desired in a product's design. Instead, it is usually more acceptable to have a trade-off between these two factors. (This essentially means the user's true preferences are nonlinear, but we show that LS is sufficient to train the policy to match any desired value function characterized by the nonlinear preference.)
>
>
> [1] Vamplew, P., Yearwood, J., Dazeley, R., Berry, A. (2008). On the Limitations of Scalarisation for Multi-objective Reinforcement Learning of Pareto Fronts
>
>
> **Q3:** *Maybe derived from a different perspective, based on Eq (9), the proposed algorithm CAPQL is almost the same as entropy-regulated RL algorithms. Is this correct?*
>
> Note: Eq (9) is now Eq (10) in the updated text.
>
> **A3:** Yes, this is correct. If we restrict ourselves to any projection of the objective functions, then they are identical. However, if a different strongly concave function is used to augment the reward, then they cease being identical. We chose entropy because of its nice numerical properties and its precedent of use. More details on its selection can be found in Appx E.
>
> **Q4**: At the bottom of page 6, g_s is not defined.
>
> **A4:** We have explicitly presented its expression in (9) in the revised version.

---

> > ### Comment · Reviewer_gKVf · 2022-11-19
> > **Response to authors' clarifications**
> >
> > I appreciate authors' helpful clarifications and modifications to the paper. I don't have further questions.

---

### Official Review · Reviewer_Z4Pt · 2022-10-26

**Confidence:** 3
**Correctness:** 4
**Technical Novelty And Significance:** 3
**Empirical Novelty And Significance:** 3
**Recommendation:** 6

**Clarity, Quality, Novelty And Reproducibility:**

This paper is clearly written. The analysis for three types of Pareto optimality for multi-objective reinforcement learning (MORL) is novel. The authors illustrate the implementation details for the experiments, but do not provide the code.

**Strength And Weaknesses:**

Strengths:
1.	This paper is very well-written and easy to follow. The authors use an example (Example 1) throughout the paper to illustrate their insights, which facilitate well readers’ understanding.
2.	The investigation on the relationship between three types of Pareto optimality is novel and interesting, which is supported by theoretical analysis.

Weaknesses:
1.	According to the results in Section 4, i.e., adding a strongly concave term can fix the problems of existing LS-based algorithms, it seems that the proposed algorithm CAPQL can add an arbitrary strongly concave term to the immediate reward. Why does CAPQL use the entropy operator $\mathcal{H}(q)$? Can this entropy operator be changed to other strongly concave terms? It would be better if the authors can include more discussion on the strongly concave terms used for CAPQL.
2.	The theoretical analysis for algorithm CAPQL is not sufficient. In the current version of this paper, the theoretical guarantee for CAPQL fully replies on the analysis for the limitations of existing LS-based algorithms and three types of Pareto optimality. It would be better if the authors can provide a more thorough theoretical analysis for CAPQL, e.g., a convergence or sample complexity guarantee.
3.	Can the authors provide the experimental results for the comparison to existing LS-based algorithms? The results in Section 4 will be more convincing if the authors can show some experimental phenomena that support the theoretical results.


**Summary Of The Paper:**

This paper studies the fundamental properties of the learning spaces in multi-objective reinforcement learning (MORL). The authors give a theoretical analysis of policy induced value functions and discuss three metrics of Pareto optimality. Their results imply the convexity of the induced value function, and show that training a policy based on linear scalarization (LS) can achieve any point of the Pareto front. The authors propose a new vector reward-based Q-learning algorithm, called CAPQL, which adds a strongly concave term to the immediate reward and solves the problems of existing LS-based algorithms.

**Summary Of The Review:**

I think that the theoretical analysis for the problems of existing LS-based algorithms and the relationship among three types of Pareto optimality for multi-objective reinforcement learning (MORL) is novel and interesting. But the proposed algorithm CAPQL lacks a sufficient discussion on the used concave term and a more in-depth theoretical analysis, e.g., a convergence or sample complexity guarantee. The experiments can be further enhanced by adding a comparison to existing LS-based algorithms and a discussion on how the empirical phenomena support the theoretical fundings in Section 4. Overall, I give borderline acceptance.

---

> ### Author Response · Authors · 2022-11-12
> **Clarifications**
>
> Thank you for reviewing our paper and providing us with feedback. We are glad that you found our theoretical analysis and notions of Pareto efficiency useful. In the discussion below, we seek to address your concerns.
>
> **Q1:** *1. According to the results in Section 4, i.e., adding a strongly concave term can fix the problems of existing LS-based algorithms, it seems that the proposed algorithm CAPQL can add an arbitrary strongly concave term to the immediate reward. Why does CAPQL use the entropy operator $\mathcal{H}(q)$? Can this entropy operator be changed to other strongly concave terms? It would be better if the authors can include more discussion on the strongly concave terms used for CAPQL.*
>
> **A1:** Yes, any other strongly concave terms can be added, but they may be difficult to use in practice. We chose to use entropy because it yields a nice solution to Eq. (10) (We have added Appx E to clarify this) and there is some precedent of entropy being used in this area (cf. our response to Q1 for ahKT).
>
> **Q2:** *The theoretical analysis for algorithm CAPQL is not sufficient. In the current version of this paper, the theoretical guarantee for CAPQL fully replies on the analysis for the limitations of existing LS-based algorithms and three types of Pareto optimality. It would be better if the authors can provide a more thorough theoretical analysis for CAPQL, e.g., a convergence or sample complexity guarantee.*
>
> **A2:** As mentioned in the main text, our algorithm, particularly when we are using entropy as our concave augmentation term, is an extension of SAC [1]. In fact, for any fixed $\mathbf{w}$, entropy CAPQL's objective function defined in (10) coincides with SAC. Therefore, in our implementation, if we focus on a specific $\mathbf{w}$, the optimization of the policy and the Q-networks is the same as the SAC's. Since Haarnoja et al. have shown that SAC converges to the optimal, CAPQL must also converge to the optimal conditioned on $\mathbf{w}$ and thus converges to the optimal for all $\mathbf{w}$ in the weight distribution's support. We have added Appx G in the revised version to discuss CAPQL's convergence property.
>
> [1] Tuomas Haarnoja, Aurick Zhou, Pieter Abbeel, and Sergey Levine. *Soft actor-critic: Off-policy maximum entropy deep reinforcement learning with a stochastic actor.* ICML, 2018
>
> **Q3:**  Can the authors provide the experimental results for the comparison to existing LS-based algorithms?
>
> **A3:** Actually, QEnv and MOQ are the two most popular LS-based methods for the MORL problem with agnostic weight preference. We have rewritten the corresponding descriptions in Sec 5.3 to clarify this.
>
> **Q4:** The results in Section 4 will be more convincing if the authors can show some experimental phenomena that support the theoretical results.
>
> **A4:** While the empirical results presented in Section 5.3 mainly show that adding the strongly concave term fixes the problems of the existing LS-based methods, we would like to note that most of the other results presented in Section 4 are supported by the numerical study. We list the corresponding empirical results below of each proposition for your reference:
>
> - Prop 3 can be seen from Figures 2, 3, and 4 as we visualized the induced function's range for Ex 1. (We want to note that for propositions like this, empirical study and verification are only feasible on toy examples. Mathematical proofs are still the only way to see why the propositions hold in general.)
>
> - Prop 4 comes directly from the convexity result (Prop 3). An intuition about this result can be built by considering any convex set and how the values on its surface align with a vector compared to those in its interior.
>
> - Prop 5 can be intuited from Fig 7 and the discussions are provided.
>
> - To show that adding a strongly concave term fixes the existing LS-based methods' problems (also discussed in Rmk 4), we compare the performance among the existing LS-based algorithm and CAPQL in Section 5.3 (c.f., our response **A3**). Additionally, we have added Appx H to visualize the estimated $g_{\mathbf w}$ found by CAPQL trained on Ex1. We believe the new plots can provide a more straightforward way to see how adding a strongly concave term helps find all PE policies and improve the numerical stability.
>
> - Rmk 5 is supported by our empirical study in Section 5.3 (*The relationship between the augmentation strength and CAPQL’s performance*).
>
> - Prop 6 and 7 are natural corollaries of Prop 4 (as mentioned in the manuscript). As a result, we do not provide extra empirical demonstrations for them.
>
> - Prop 8 is supported by the discussions in Section 3.2, which contains a detailed empirical study.
>
> **Remark:** *The authors illustrate the implementation details for the experiments, but do not provide the code.*
>
> **A:** We have now included a link to our implementation of CAPQL and QEnv_ctn in our post at the top.

---

> > ### Comment · Reviewer_Z4Pt · 2022-11-20
> > **Thanks for the rebuttal**
> >
> > Thank you for the detailed reply and new experimental results. My concerns are addressed. I will keep my score.

---

### Official Review · Reviewer_HSHv · 2022-10-28

**Confidence:** 4
**Clarity, Quality, Novelty And Reproducibility:** The algorithm table and the correspon…
**Correctness:** 3
**Technical Novelty And Significance:** 2
**Empirical Novelty And Significance:** 2
**Recommendation:** 5

**Strength And Weaknesses:**

The paper analyses the dynamics of the induced value functions resultant from policy alterations in MORL problems. The analysis seems good.

The proposed algorithm mainly adds a concave function of the policy, which can help with determinism and numerical instability. The algorithms could have some sample complexity type results which will be helpful, with comparisons to the existing works. This is especially since many algorithms like policy gradient, actor critic, etc. have sample complexity guarantees. Even though this paper is different from CURL frameworks, it is not clear why we cannot use the model like there to model the multi-objective problem - and what benefits this formulation gives or what problems this could solve which the other cannot.

**Summary Of The Paper:**

This paper discusses the impact of multiple objectives. Pareto optimality guarantees are discussed, and actor-critic formulation is provided.

**Summary Of The Review:**

The paper provides some results on Pareto optimality of MORL with concave regularization of the policy. Sample complexity results would have been nice for the paper.

---

> ### Author Response · Authors · 2022-11-12
> **Clarifications (part 1)**
>
> Thank you for taking the time to review our paper. Our responses seek to clarify how our work is different from Agarwal et al.'s results and what we have done in the revised version to avoid potential confusion.
>
> **Q1:** *The key negatives is that the reviewer is not finding the key new result with the existing literature on MORL. See for instance "Multi-Objective Reinforcement Learning with Non-Linear Scalarization" in AAMAS 2022. The Pareto properties of MORL have been shown, and the actor critic algorithm is also provided. Thus, two of the novelties mentioned in this paper already exist unless the reviewer is missing something.*
>
> **A1:** We do believe that there is a misunderstanding. Our work is in fact very different from Agarwal et al.'s.
>
> First of all, the two works have different focuses. One important part of our work is to investigate the properties of the induced value function's range and discuss the relationship among three different PE policies. These insights explain why the linear scalarization cannot find all PE policies (up to the same induced value functions) and motivates us to add a strongly concave term to fix the problems. In contrast, Agarwal et al.'s work does not provide similar results and directly starts by proposing a concave scalarization-based framework.
>
> Secondly, our work is based on a different problem configuration and has different assumptions. Our configuration considers the scalarization of the immediate rewards, while Agarwal et al. apply the scalarization over the sum of discounted rewards. The assumptions on the scalarization functions are also different. In Agarwal et al.'s work, the function is assumed to be L-Lipschitz concave (and element-wise monotone to ensure the found policy is PE). In contrast, our CAPQL algorithm simply uses linear scalarization (and adds a strongly concave term to the immediate reward).
>
> Additionally, the policies produced by the two works are different. Agarwal et al.'s framework typically finds one PE policy while CAPQL is designed to find all the PE policies (up to the same induced value functions) that cover the entire Pareto front of the induced value function's range.
>
> Moreover, for Agarwal et al.'s framework, the selection of the concave scalarization function is task-dependent. However, in CAPQL, the added strongly concave term is to make the function $g_s$ given in (9) well-defined, surjective and uniformly continuous. Therefore, its selection mainly depends on its computational efficiency (see Appx E) and can be shared by different tasks.  (The CAPQL algorithm evaluated in this paper is induced by setting the concave term to the entropy function $\mathcal{H}$.)
>
> We disagree that our work reproduces known Pareto properties. Pareto optimality has been discussed at length in many previous works, but none of these focus on our problem setting. Agarwal et al. do not define different classes of Pareto efficiency, nor do they analyze the set of value functions for convexity.
>
> Likewise, Actor-Critic is a general class of algorithms, but the details of the algorithms can vary significantly. It appears that the CALQL only shares the most general framework with their paper (having a separate memory structure to explicitly represent the policy independent of the value or Q-networks). However, our methods are fundamentally different and solve different problems. If you have a specific concern about the similarity here, we would be happy to clarify.
>
> We believe that our work does not repeat Agarwal et al.'s work and in fact considers a different problem, a different algorithm and includes novel insights. We hope that the discussion above has clarified our contributions and we are happy to make further clarification if you have any specific concerns regarding their similarity.
>
> In the revised version, we have differentiated our contributions from Agarwal et al.'s paper in the related works section to avoid potential confusion.

---

> > ### Author Response · Authors · 2022-11-12
> > **Clarifications (part 2)**
> >
> >
> > **Q2:** *A better comparison to the MORL literature and more generally CURL literature should be provided, and the new results should be emphasized. Also, the algorithms could have some sample complexity type results which will be helpful, with comparisons to the existing works.*
> >
> > **A2:** As mentioned in our previous response, our work is quite different from the one presented in "Multi-Objective Reinforcement Learning with Non-Linear Scalarization"; therefore, we do not see a significant benefit to reviewing CURL literature in detail. Instead, in our revised version, we have briefly mentioned this broader field for potential readers' interests.
> >
> > Regarding the sample complexity, we believe our work is more related to QENV [1], MOQ [2], SAC [3] (and we made empirical comparisons with them). None of the works provide sample complexity results. We are open to further discussion on this point, but we think this could be a digression and will not significantly enhance our contributions.
> >
> > [1] Runzhe Yang, Xingyuan Sun, and Karthik Narasimhan. A generalized algorithm for multi-objective reinforcement learning and policy adaptation. NIPS, 2019.
> >
> > [2] Axel Abels, Diederik M. Roijers, Tom Lenaerts, Ann Nowe, and Denis Steckelmacher. *Dynamic weights in multi-objective deep reinforcement learning.* ICML, 2019.
> >
> > [3] Tuomas Haarnoja, Aurick Zhou, Pieter Abbeel, and Sergey Levine. *Soft actor-critic: Off-policy maximum entropy deep reinforcement learning with a stochastic actor.* ICML, 2018.

---

> ### Author Response · Authors · 2022-11-21
> **Further Clarifications**
>
> We are glad that we were able to address your concerns about the novelty.
>
> *For the problem about sample complexity analysis*, we would like to note that a large portion of our theoretical work is to show LS is provably sufficient to find all PE policies (up to the same value functions) and explain why the existing algorithms fail. We then show that the failures of LS can be avoided by adding a strongly concave term to the immediate reward. The CAPQL algorithm follows this theoretical guidance and has been shown to perform constantly better than the existing LS algorithms, which in turn corroborates our theory. While we believe that sample complexity analysis is an interesting topic, we are not sure how including it in this paper would make the entire package stronger. Additionally, to the best of our knowledge, we are not aware of any papers in preference agnostic MORL settings that provide sample efficiency bounds. If you have any specific references in this domain you wish us to compare to, we are willing to take a look and have a further discussion.
>
> *Regarding CURL frameworks*, our paper has never hinted that the formulation used in CURL was wrong or inappropriate. Specifically, in our paper, we used the most standard RL formulation and revealed some good properties that were not believed true in general; however, this does not mean that CURL's formulation is useless and we will make this clear in the final version.

---

### Official Review · Reviewer_ahKt · 2022-11-01

**Confidence:** 3
**Correctness:** 4
**Technical Novelty And Significance:** 3
**Empirical Novelty And Significance:** 3
**Recommendation:** 6

**Clarity, Quality, Novelty And Reproducibility:**

The paper is well-written and relatively easy to follow. The theoretical novelty is high and relevant to the multi-objective RL community.

**Strength And Weaknesses:**

Strength: overall the theoretical contribution is quite interesting and insightful. As the authors pointed out, recent multi-objective RL research has not presented a coherent view on this topic. This paper is a timely investigation of some fundamental properties in this space.

Weakness:
The empirical evaluation section, in comparison, is weak. As the new algorithm proposes adding a strongly concave term to the reward function, it would be better to explain the choice of selecting the entropy operator in the implementation. Can one use any other operator in its place?

While the results on the 4 presented environments look promising, I noticed the sample size is relatively small (10k steps). Are there longer runs so we can see result comparison at convergence?

The authors should also discuss a relevant paper on introducing an improved scalarization method beyond linear scalarization.

Kristof Van Moffaert; Madalina M. Drugan; Ann Nowé: Scalarized multi-objective reinforcement learning: Novel design techniques


**Summary Of The Paper:**

This paper studies some theoretical questions about multi-objective reinforcement learning. The focus is on policy-induced value functions. Section 3 develops several tools to characterize changes to value functions when one changes policies. Section 4 uses those tools to show the value function space is convex and provides insights into why linear scalarization methods fail to find Pareto-efficient policies. The authors present a concave-augmented Pareto Q-learning algorithm (CAPQL) and demonstrate its effectiveness on several multi-objective variants of the MuJoCo environments.

**Summary Of The Review:**

This a well-written paper with a good theoretical contribution. The empirical evaluation section could be strengthened to make the whole package stronger.

---

> ### Author Response · Authors · 2022-11-12
> **Clarifications.**
>
> We thank you for your time and comments. We are glad that you found the paper interesting and overall worthwhile. We have given the following responses/explanations to clarify aspects of our paper.
>
> **Q1:** *The empirical evaluation section, in comparison, is weak. As the new algorithm proposes adding a strongly concave term to the reward function, it would be better to explain the choice of selecting the entropy operator in the implementation. Can one use any other operator in its place?*
>
> **A1:** In theory, any other strongly concave policy function can be used. There were two main motivations for selecting entropy as the concave term: precedent and mathematical convenience. Several previous works have used entropy maximization as a regularization term and achieved good results [1-5]. Mathematically, using entropy allows us to have a computationally efficient solution to Eq. (10). For more details, please see the newly created Appendix E. We have updated the main text of the paper to refer to this appendix.
>
> [1] Tuomas Haarnoja et al., Soft Actor-Critic: Off-Policy Maximum Entropy Deep Reinforcement Learning with a Stochastic Actor
>
> [2] Ziebart, B. D., Maas, A. L., Bagnell, J. A., and Dey, A. K.Maximum entropy inverse reinforcement learning.   In AAAI Conference on Artificial Intelligence (AAAI), pp.1433–1438, 2008.
>
> [3] Ziebart, B. D.Modeling purposeful adaptive behavior with the principle of maximum causal entropy. Carnegie Mellon University, 2010.
>
> [4] Rawlik, K., Toussaint, M., and Vijayakumar, S. On stochastic optimal control and reinforcement learning by approximate inference.Robotics: Science and Systems (RSS),2012.
>
> [5] Nachum,  O.,  Norouzi,  M.,  Xu,  K.,  and Schuurmans,  D.Trust-PCL: An off-policy trust region method for continuous control.arXiv preprint arXiv:1707.01891, 2017b.
>
> **Q2:** *While the results on the 4 presented environments look promising, I noticed the sample size is relatively small (10k steps). Are there longer runs so we can see result comparison at convergence?*
>
> **A2:** The runs are in fact far longer than 10k. What we meant by "10k" was that the x-axis is in units of 10k steps. Thus our experiments run close to 1,000,000 steps. We have updated Fig 8 and Fig 9 to avoid any potential confusion.
>
> **Q3:** *The authors should also discuss a relevant paper on introducing an improved scalarization method beyond linear scalarization.
> Kristof Van Moffaert; Madalina M. Drugan; Ann Nowé: Scalarized multi-objective reinforcement learning: Novel design techniques*
>
> **A3:** Thank you for the suggestion. We have added a small discussion of its contributions in our related works section (Appendix A).

---

### Author Response · Authors · 2022-11-17
**General Reminder of Manuscript Changes Deadline**

We just wanted to send out a quick reminder that tomorrow (November 18th) is the last day that we can update our manuscript, so if there are any more last-minute changes that you would like us to make/include in our paper, please let us know.

---

### Author Response · Authors · 2022-11-22
**Summary of our dialogue with the reviewers**

We thank the reviewers for their time and constructive feedback. We would like to summarize the responses the reviewers have given us and how we have addressed them.

### Strengths
- The theoretical contributions are interesting and insightful. (**ahKt**)
- The theoretical results are novel. (**ahKt**, **gKVf**)
- The theoretical work on the three types of PE are novel and interesting. (**Z4Pt**, **gKVf**)
- The analysis is rigorous. (**gKVf**)
- The theoretical analysis looks good. (**HSHv**)
- The contributions will have a large impact on the MORL community. (**gKVf**)
- The contribution is timely. (**ahKt**)
- The paper is well-written and easy to follow. (**ahKt**, **Z4Pt**, **gKVf**)

### Concerns

- [**Resolved**] The work should discuss the selection of the strongly concave term used for CAPQL. (**ahKt**, **Z4Pt**)
    - We added Appx E to discuss the selection, and **Z4Pt** has confirmed their satisfaction.
- [**Resolved**] Empirical study should have a larger sample size. (**ahKt**)
    - The sample size was much larger than what was thought, and we have modified Fig 8 to avoid confusion.
- [**Resolved**] The key concern is: the work repeats "Multi-Objective Reinforcement Learning with Non-Linear Scalarization" in AAMAS2022. (**HSHv**)
    - We made a detailed comparison in our responses and modified our related work to avoid confusion. This concern appears to have been resolved as the related sentences have been removed after the rebuttal.
- [**Resolved**] The authors should discuss the theoretical guarantees of CAPQL. (**Z4Pt**)
    - Appx G has been added to discuss the theoretical guarantees of CAPQL, and **Z4Pt** has confirmed their satisfaction.
- [**Resolved**] The authors should compare their algorithm with the existing LS-based algorithms. (**Z4Pt**)
    - We modified the manuscript to clarify that QEnv and MOQ are the most popular LS_based algorithms, and **Z4Pt** has confirmed their satisfaction.
- [**Resolved**] The work should discuss how the empirical phenomena support the theoretical findings in Section 4. (**Z4Pt**)
    - We explicitly draw the connections between our theoretical works and numerical results in the responses and provided additional results in Appx H. **Z4Pt** has confirmed their satisfaction.
- [**On-going**] It is unclear why we should not use the CURL framework. (**HSHv**)
    - We note that our work does not suggest the CURL framework is inferior or inappropriate. We are willing to further clarify this if needed.
- [**On-going**] The work could have complexity type results. (**HSHv**)
    - While Appx G has been added to discuss the theoretical guarantees, the authors still believe a stronger version (like sample complexity analysis) does little to enhance the current submission.

We believe most of the concerns have been resolved, and we are happy to engage in further discussions and provide additional clarifications/comments.

---

### Author Response · Authors · 2022-11-25
**Thank you for your time and effort**

We would like to thank the ACs and reviewers for taking the time to review our paper and give thoughtful responses. The feedback we have received during this time has helped us increase the quality of our paper and we are grateful for this. If there are any unaddressed concerns, we are happy to clarify.

---

### Decision · Program_Chairs · 2023-01-20

**Decision:**

Accept: poster

**Justification For Why Not Higher Score:**

The paper has some limitations on the empirical analysis.

**Justification For Why Not Lower Score:**

The paper addresses a relevant topic and introduces interesting contributions.
The limitations of the paper are not so significant to prevent publication.

**Metareview: Summary, Strengths And Weaknesses:**

The paper studies the fundamental properties of the policy-induced value functions in multi-objective reinforcement learning (MORL).
The authors identify issues in training a policy via linear scalarization of the objectives and propose to fix them by adding strongly concave terms to the immediate rewards. The proposed approach has shown to be more stable in solving multiple MuJoCo tasks.
After reading each others' reviews and the authors' feedback, the reviewers start to discuss the strong and weak points of this paper.
They appreciated the theoretical contributions, while there were some concerns about the empirical validation, such as the realization of ablation studies.
Overall, during the discussion, the reviewers agreed that the positive aspects outweighed the negative ones.
The authors must follow the reviewers' suggestions while preparing their camera ready.

**Note From Pc:**

if the above contains the word "oral" or "spotlight" please see: "oral" presentation means -> notable-top-5% and "spotlight" means -> notable-top-25%. As stated in our emails, we are disassociating presentation type from AC recommendations

**Summary Of Ac-Reviewer Meeting:**

Only two reviewers participated in the virtual meeting (Reviewer HSHv and Reviewer ahKt).
The two reviewers agreed that the paper is borderline, but after discussing their concerns and the other reviewers' concerns (all the issues are mainly focused on the empirical evaluation), they agreed that some of these issues could be easily addressed by the authors in the final version, and the other issues do not prevent publication.
For these reasons, I decided to accept the paper.